# Genetically controlled membrane synthesis in liposomes

Duco Blanken[1,2], David Foschepoth [1,2], Adriana Calaça Serrão[1] & Christophe Danelon[1✉]

Lipid membranes, nucleic acids, proteins, and metabolism are essential for modern cellular life. Synthetic systems emulating the fundamental properties of living cells must therefore be built upon these functional elements. In this work, phospholipid-producing enzymes encoded in a synthetic minigenome are cell-free expressed within liposome compartments. The de novo synthesized metabolic pathway converts precursors into a variety of lipids, including the constituents of the parental liposome. Balanced production of phosphatidylethanolamine and phosphatidylglycerol is realized, owing to transcriptional regulation of the activity of specific genes combined with a metabolic feedback mechanism. Fluorescence-based methods are developed to image the synthesis and membrane incorporation of phosphatidylserine at the single liposome level. Our results provide experimental evidence for DNA-programmed membrane synthesis in a minimal cell model. Strategies are discussed to alleviate current limitations toward effective liposome growth and self-reproduction.

[1] Department of Bionanoscience, Kavli Institute of Nanoscience, Delft University of Technology, Van der Maasweg 9, 2629 HZ Delft, The Netherlands. [2]These authors contributed equally: Duco Blanken, David Foschepoth. ✉email: c.j.a.danelon@tudelft.nl

Biological cells are spatially delimited from their surrounding by a lipid membrane. While archaeal membranes are composed of ether lipids, other cell types use phospholipids as the most abundant membrane constituents. Most phospholipids self-assemble in aqueous solutions to form vesicles, called liposomes, under a wide range of experimental conditions. Spatial organization of biochemical processes within liposomes mimics the fundamental characteristics according to which natural cells are organized. Therefore, phospholipid vesicles provide a chassis for the construction of synthetic minimal cells representing comparatively simple model systems[1–5].

Also pertinent to a working definition of cellular life is the notion of self-maintenance, in line with the view of a basic cell as an autopoietic unit[6], whereby all the system's components are produced within its boundary. Substrates present in the external environment absorb to the membrane or diffuse across, and are transformed into molecular building blocks by metabolic processes. Another aspect that is particularly relevant when describing the inner functioning of a biological cell is the coupling between the different subsystems[7], such as genetic information, protein synthesis, and metabolic synthesis of the membrane constituents. Herein, we apply this conceptual framework to the construction of a minimal cell that can produce its own membrane components. Cell-free protein and phospholipid synthesis, directed by a DNA program, is carried out inside a liposome, constituting an integrative step on the way to the development of an autonomously growing and dividing artificial cell.

Various strategies have been described to grow liposomes. Membrane constituents directly supplied in the external medium in the form of monomers, micelles or small unilamellar vesicles can spontaneously adsorb or fuse to the liposome membrane, increasing its surface area[8–11]. Moreover, non-enzymatic mechanisms to produce membrane lipids from synthetic reactive precursors and catalysts are particularly effective, leading to substantial vesicle growth[12–15]. To establish a link between the lipid compartment and its internal content, liposome growth could be made conditional to encapsulated nucleic acids[12,16] or catalysts[17]. Such model systems are attractive for their molecular simplicity and may resemble primitive cells before the emergence of modern biology. Closer to processes occurring in contemporary cells, enzyme-catalysed biosynthesis of phospholipids has been realised using purified proteins[18–21]. Further, the lipid-producing enzymes were encoded in DNA and expressed by in vitro protein synthesis inside liposomes, providing a genotype-to-phenotype linkage[22,23]. The Escherichia coli enzymes glycerol-3-phosphate (G3P) acyl transferase and lysophosphatidic acid (LPA) acyl transferase, respectively referred below as PlsB and PlsC from their gene names, were in situ expressed from two DNA templates[23]. The precursors G3P and fatty acyl coenzyme A (acyl-CoA) were sequentially converted into lysophosphatidic acid and phosphatidic acid (PA) lipids in a two-step enzymatic reaction (Fig. 1a). However, the output phospholipid PA was not part of the original membrane composition and the PA detection method was not compatible with single vesicle resolution[23]. Regeneration of the main constituents of the liposome membrane obligates the reconstitution of five additional headgroup-modifying enzymes, which together with PlsB and PlsC form the Kennedy metabolic pathway that produces phosphatidyl-ethanolamine (PE) and phosphatidylglycerol (PG), the most abundant lipids in the E. coli membranes. Although the unregulated expression of the Kennedy pathway enzymes was enabled from the outside of liposomes[23], in vesiculo synthesis of membrane-forming lipids with controlled molecular ratios remains a challenge.

In the present work, we show that the synthesis of PE and PG lipids from simpler precursors can be genetically controlled inside PE- and PG-containing liposomes. Our results provide experimental evidence for DNA-encoded membrane synthesis in a liposome-based artificial cell. Because the metabolic pathway encompasses seven different enzymes, we first assemble all seven genes on a single plasmid. The PURE (Protein synthesis Using Recombinant Elements) system[24], here PUREfrex2.0, is used as a minimal cell-free protein synthesis platform that converts the DNA program into the whole enzymatic pathway. Phospholipid biosynthesis within liposomes is demonstrated by quantitative liquid chromatography-mass spectrometry (LC-MS). Relative PE and PG content is tailored through transcriptional and metabolic regulation mechanisms. Moreover, we develop fluorescence-based probes to directly visualize membrane incorporation of synthesized phospholipids at the single vesicle level.

## Results

**Design of a minigenome for phospholipid biosynthesis**. We aimed to reconstitute the Kennedy phospholipid synthesis pathway from E. coli starting from all seven enzyme-encoded genes (Fig. 1a). The membrane-bound protein PlsB uses acyl-CoA (or acyl carrier protein, ACP) as a donor to acylate the 1-position of G3P to form LPA[25]. The 2-position is subsequently acylated by the membrane protein PlsC to form diacyl PA, again using acyl-CoA as fatty acid donor, preferring unsaturated carbon chains[25]. Enzymes downstream in the pathway are involved in phospholipid headgroup modifications. The integral membrane protein CdsA catalyses the activation of PA with cytosine triphosphate (CTP) to generate diacyl-sn-glycero-3-(cytidine diphosphate) (diacyl-CDP-DAG)[26] which serves as a precursor for two separate branches of the Kennedy pathway. One branch, which leads to the formation of PG as the final product, comprises the synthesis of phosphatidylglycerol phosphate (PGP) from G3P and CDP-DAG by the membrane-associated protein CDP-diacylglycerol-glycerol-3-phosphate 3-phosphatidyltransferase (PgsA)[27], followed by a dephosphorylation step that is catalysed by the phosphatidylglycerophosphatase A, B[28] or C (PgpA, B or C)[29]. The other branch generates PE as the end-product in a two-step reaction. First, phosphatidylserine (PS) production from CDP-DAG and L-serine is catalysed by the CDP-diacylglycerol-serine O-phosphatidyltransferase (PssA). Then, PS is decarboxylated to form PE, a reaction that is catalysed by the phosphatidylserine decarboxylase (Psd), a two-subunit protein resulting from the autocatalytic serinolysis of a single proenzyme[30].

All seven genes, namely plsB, plsC, cdsA, pgsA, pgpA pssA, and psd were concatenated into a single plasmid DNA as individual transcriptional cassettes, i.e. every gene is under control of its own promoter, ribosome binding site and transcription terminator (Fig. 1a). This design strategy ensures that all genes will be present at the same copy number upon plasmid encapsulation in liposomes, thus obviating the functional heterogeneity inherent to uneven partitioning of the separate DNA templates. Thirty base pair linker sequences were added to each gene and to a linearized pUC19 plasmid backbone by polymerase chain reaction (PCR) to enable a one-step Gibson-assembly of the final plasmid[31] (Supplementary Figs. 1 and 2). The successful assembly of the pGEMM7 minigenome was confirmed using Sanger sequencing and restriction digestion (Supplementary Fig. 3). The three genes of the common pathway plsB, plsC and cdsA, as well as the two genes pssA and psd of the PE synthesis branch are under control of a T7 promoter and are constitutively expressed in PUREfrex2.0. The two genes pgsA and pgpA, encoding the enzymes for PG biosynthesis, are under control of an SP6 promoter and are encoded on the opposite strand to prevent read-through transcription by incomplete termination at the T7 terminator sites (Supplementary Note 1, Supplementary Fig. 4).

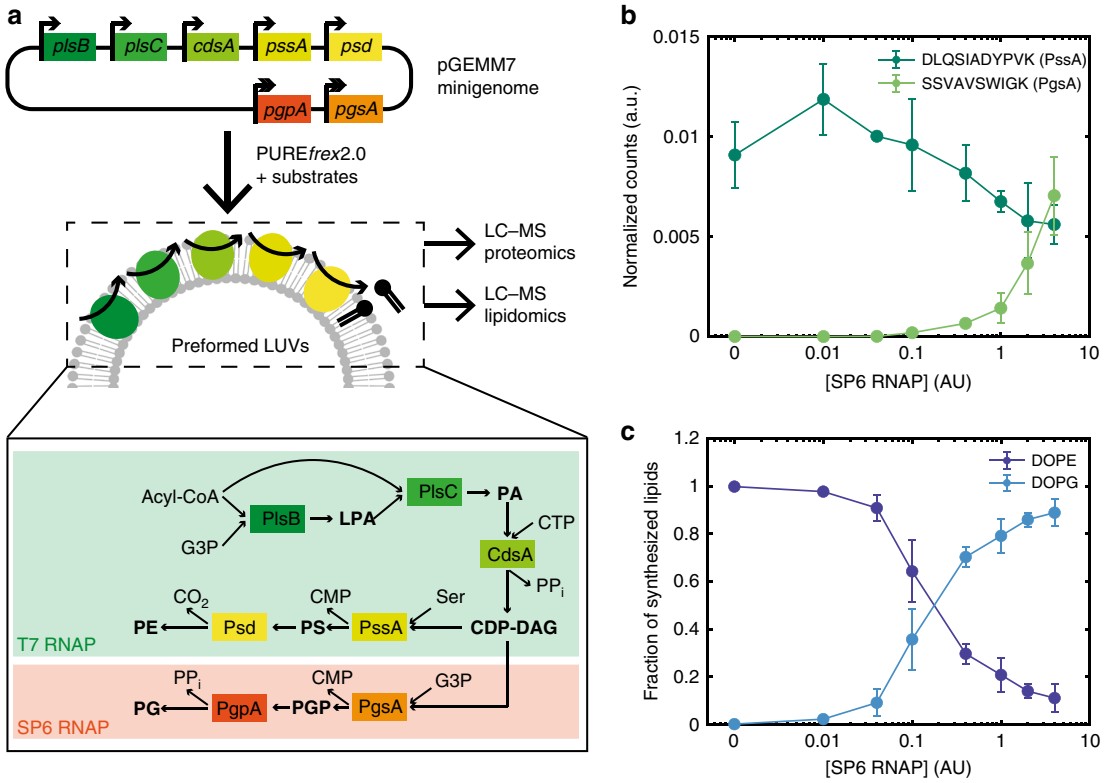

**Fig. 1 Genetically controlled production of PE and PG by de novo synthesized enzymes. a** The pGEMM7 plasmid contains seven genes encoding *E. coli* lipid synthesis enzymes. Transcriptional regulation over the production of PE and PG lipids is provided by controlling the expression of specific enzymes with the orthogonal T7 (single arrow) and SP6 (double arrow) promoters. The main reaction products are in bold and the enzyme names are squared. The PgsA–PgpA branch of the pathway, which leads to PG synthesis, is activated upon addition of the SP6 RNAP. An acyl-CoA, the heavy isotope of G3P, CTP, and serine (Ser) are the input substrates. Expression of pGEMM7 with PURE*frex*2.0 occurred in the presence of preformed LUVs. Protein and lipid production was monitored by LC-MS. PPi, pyrophosphate. **b** LC-MS analysis of cell-free synthesized proteins. Normalised integrated peak intensity for representative peptides of PssA and PgsA, the first enzymes after the pathway branches out, for a range of SP6 RNAP concentrations (given in activity units as defined by the supplier, AU). Data are the mean ± SD of three independent experiments. **c** LC-MS analysis of de novo synthesized phospholipids. The fraction of synthesized DOPE and DOPG is plotted for a range of SP6 RNAP concentrations (in AU). Data are the mean ± SD of three independent experiments. Source data are available for (**b**, **c**).

Orthogonality of the two promoter-RNA polymerase (RNAP) pairs in PURE*frex*2.0 was demonstrated using a fluorescent protein reporter (Supplementary Fig. 5).

**Transcriptional regulation of PE and PG biosynthesis.** Traditionally, cell-free translation products are characterized by one-dimensional sodium dodecyl sulfate-polyacrylamide gel electrophoresis (SDS-PAGE) using isotopically or fluorescently labelled amino acids as a readout. While these methods are suitable to analyse single or a few gene expression products, they suffer from a poor resolution when multiple proteins are co-synthesized (Supplementary Fig. 6). Here, we applied a targeted LC-MS proteomics approach to detect the de novo synthesized enzymes and validate transcriptional activation of the PgsA–PgpA pathway by the SP6 RNAP.

Large unilamellar vesicles (LUVs) supplied in PURE*frex*2.0 reactions served as a scaffold for the expressed membrane-associated and integral membrane proteins. Several proteolytic peptides of the expressed proteins were identified (Supplementary Table 4, Supplementary Fig. 7) and the total ion current of their observed fragment ions was normalized to a peptide originated from elongation factor thermo unstable (EF-Tu), an abundant protein in PURE system. In-solution digestion of pre-ran PURE system reaction samples with trypsin failed to deliver detectable peptides for one of the seven proteins, namely PgpA (Supplementary Note 2). No detectable amount of PgsA was measured when the SP6 RNAP was omitted, indicating that unintended expression of the *pgsA* gene is negligible (Fig. 1b). Varying the concentration of SP6 RNAP between 0.01 U μL⁻¹ and 4 U μL⁻¹ is accompanied by a gradual increase in PgsA. Concurrently, the concentration of the PssA enzyme under T7 promoter control decreases upon increased SP6 RNAP concentration. These results show the power of targeted proteomics for relative quantification of cell-free protein synthesis. Moreover, they validate our design for tuneable expression levels of different enzymes belonging to orthogonal transcriptional pathways.

Successful production of PE and PG lipids and its genetic modulation were confirmed by an LC-MS lipidomics analysis (Fig. 1c, Supplementary Fig. 8). To distinguish the newly produced lipids from those initially present in the liposome membrane, ¹³C-labelled G3P was used as an isotopically heavy precursor. Oleoyl-CoA was used as the acyl donor. Absolute quantification was achieved by measuring DOPG and DOPE standards prior to and post data acquisition of PURE system samples. In agreement with proteomics data, synthesized DOPG was detected exclusively in the presence of SP6 RNAP (Fig. 1c). The only intermediate species that significantly accumulates is DOPA (Supplementary Fig. 9).

**Metabolic regulation of PE and PG biosynthesis.** PssA is unique among the proteins of the Kennedy pathway since it is found both associated with the membrane and in the cytosol[32,33]. PssA is

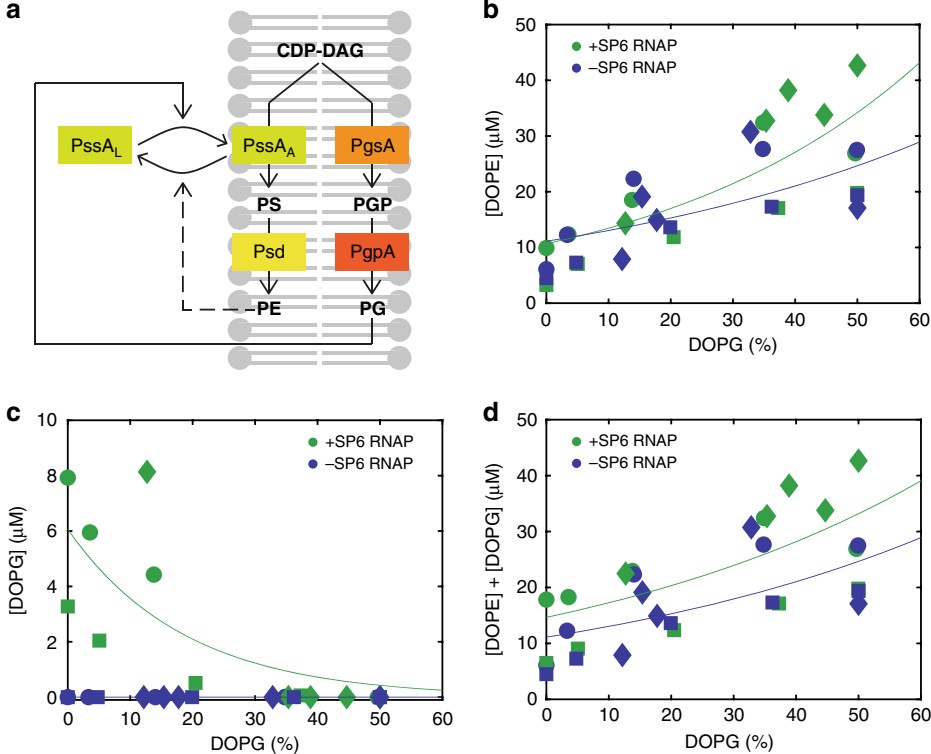

**Fig. 2 Metabolic feedback as a regulator for cell-free synthesis of PE and PG. a** Schematic illustration of PssA activity regulation by membrane content. PssA exists in the membrane-bound, activated state (PssA$_A$) and in the cytosolic, inactivated or latent state (PssA$_L$). High amounts of PG favour PssA$_A$ by promoting membrane recruitment, thus increasing the yield of synthesized PE. Low PG content (i.e. high fraction of PE) shifts the equilibrium to PssA$_L$, channelling resources to the production of PG and reducing the fraction of synthesized PE. **b–d** Concentrations of synthesized PE (**b**), PG (**c**) and PE + PG (**d**) for different initial compositions of LUVs in the presence (green symbols) or absence (blue symbols) of SP6 RNAP. The three different markers represent data from three independent experiments. The solid lines are appended to guide the eye. Membranes always contain 50 mol% DOPC and varying fractions of DOPG and DOPE. Source data are available for (**b–d**).

thought to maintain the ratio between acidic (PG and cardiolipin, CL) and zwitterionic (PE) lipids in *E. coli* by being activated upon association with PG/CL-rich membranes, whereas the cytosolic form is latent (Fig. 2a)[34,35]. We sought to exploit this feedback mechanism to provide membrane content homeostasis without relying on genetic control. LUVs with different amounts of DOPE and DOPG were prepared, and synthesis of [13]C-labelled DOPE and DOPG was determined by LC-MS, both in the presence and absence of SP6 RNAP (Fig. 2b, c). A clear positive correlation between initial PG content and yield of synthesized PE was observed, both in the presence ($\rho = 0.91 \pm 0.07$, mean ± SD of three independent repeats) and absence ($\rho = 0.94 \pm 0.04$, mean ± SD of three independent repeats) of SP6 RNAP. Moreover, a negative correlation between initial PG content and yield of synthesized PG was observed ($\rho = -0.95 \pm 0.03$, mean ± SD of three independent repeats). These results confirm the model of allosteric regulation of PssA activity by PG content, providing non-genetic homeostasis of mixed lipid composition to our system. Interestingly, PE synthesis was reduced at low PG content, independent of the expression of the PG-synthesizing pathway branch (Fig. 2b). This result indicates that the regulatory mechanism is not solely driven by competition between the two pathway branches but it relies also on the association-dissociation of PssA to the membrane (Fig. 2a)[30–32]. We also found that the total amount of synthesized PE and PG is ~2-fold higher at a higher mol% of initial PG (~18 μM at 0 mol% PG vs. ~28 μM at 35 mol% PG in the experiment shown in Fig. 2d). This result is in line with previous observations that PlsB activity is promoted by PG[36,37].

**Compartmentalised biosynthesis of PE and PG in liposomes.** Lipid synthesis localised inside individual liposomes is of paramount importance in the realization of autonomously growing artificial cells. The successful reconstitution of the seven gene-encoded enzymes for PE and PG synthesis in the presence of LUVs prompted us to confine the entire chain of reactions inside cell-sized liposomes that initially contain PE and PG lipids. PURE system, pGEMM7 minigenome and soluble phospholipid precursors were encapsulated inside large and giant liposomes. Acyl-chain precursors were supplied as a dried film and, when suspended in the aqueous solution, partitioned in the membrane of liposomes. Cell-free gene expression was restricted to the liposome lumen by adding either proteinase K or DNase I in the external medium. In-liposome gene expression was first validated using the yellow fluorescent protein (YFP) as a reporter (Fig. 3a, b). Quantitative mass spectrometry analysis of synthesized lipid products showed that it is possible to synthesise up to 20 μM of phospholipid end products, corresponding to an acyl-CoA conversion yield of 40%, when all reactions are confined to the liposome lumen (Fig. 3d). Both acyl-chain precursors palmitoyl-CoA (16:0) and oleoyl-CoA (18:1) could be used as substrates, resulting in the synthesis of dipalmitoyl and dioleoyl phospholipids, respectively (Fig. 3d, Supplementary Fig. 10). Because the newly synthesized DOPE and DOPG are also constituents of the parental liposomes, this result represents a milestone towards homeostatic membrane growth directed from genomic DNA. Control experiments without proteinase K (Fig. 3d, Supplementary Fig. 10) result in only slightly higher phospholipid yields, despite the much larger reaction volume of the extravesicular space. This could suggest a

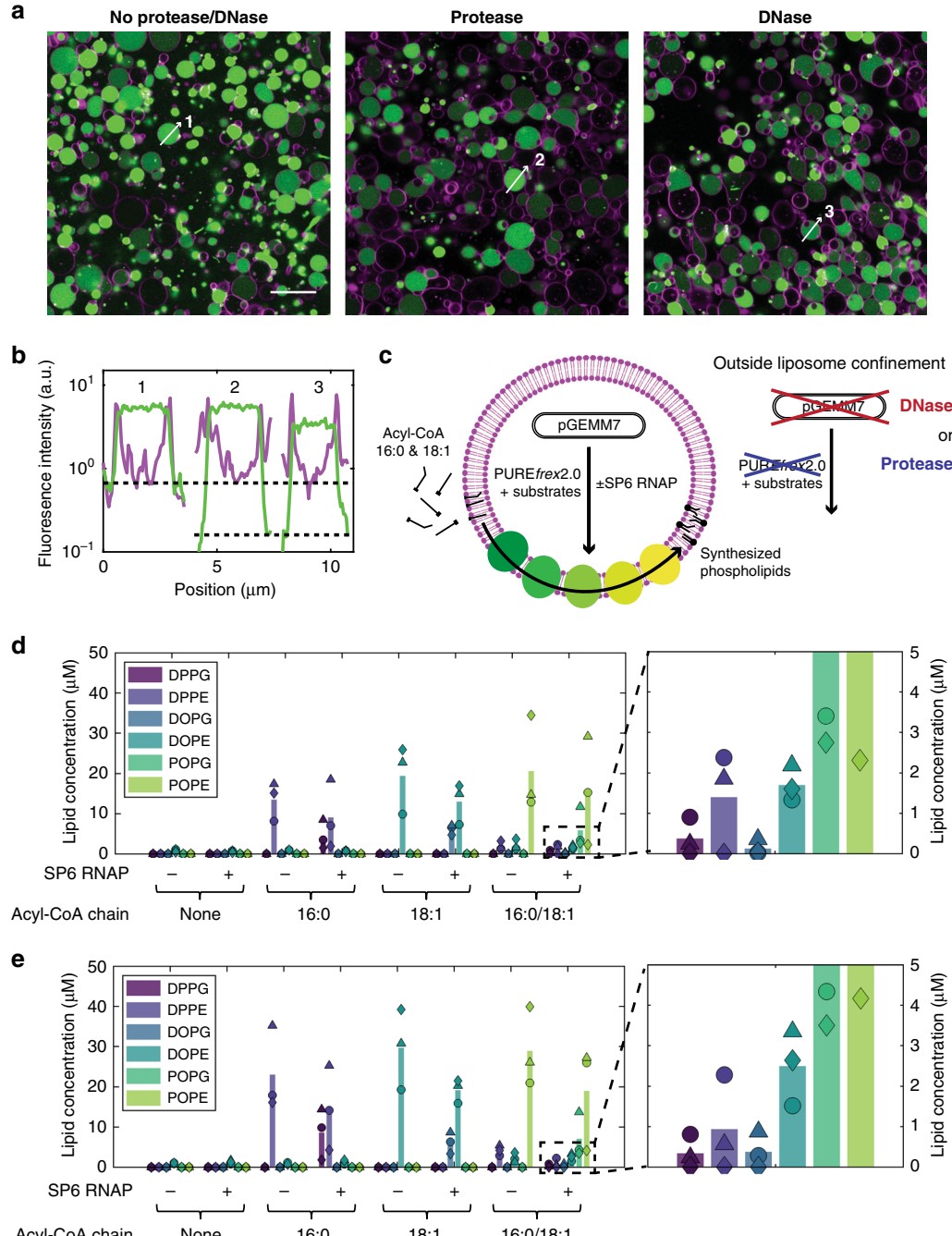

**Fig. 3 DNA-programmed phospholipid synthesis inside giant vesicles. a** Fluorescence images of liposomes (membrane in magenta) encapsulating PURE*frex*2.0, 5 mM β-mercaptoethanol and 7 nM DNA encoding for the YFP (in green). Gene expression was confined inside liposomes by external addition of either proteinase K (middle) or DNase I (right), or was allowed to also occur outside liposomes (left). Liposomes were diluted (2 μL in 7.5 μL total) to reduce their surface density and aid visualisation. Scale bar indicates 20 μm; all pictures have the same size and were acquired with identical imaging settings. This experiment was repeated twice and similar results were obtained. **b** Fluorescence intensity line profiles for the liposomes indicated in (**a**). Colour coding is the same as in (**a**). Dotted black lines indicate the background YFP level without (top) and with (bottom) protease/DNase. **c** Schematic representation of in-liposome gene expression coupled to phospholipid synthesis. **d, e** Concentration of synthesized phospholipids inside (**d**) or both inside and outside liposomes (**e**), as determined by LC-MS. Compartmentalisation of gene expression was ensured by addition of proteinase K in the external environment. Different combinations of acyl-CoA precursors and SP6 RNAP were used. Concentrations of oleoyl-CoA and palmitoyl-CoA were 100 μM when added separately, and 50 μM each when added together. Symbols indicate measurements from three independent experiments and the bars represent mean values. A small amount of DOPE was measured in samples where no acyl-CoA was supplied. This represents the naturally occurring heavy-isotope fraction of the DOPE contained in the initial liposome membrane. The right panels are blow-up graphs of the indicated area in the left panels. Source data are available for (**d**, **e**).

possible enhancement of gene expression and/or lipid synthesis by encapsulation inside liposomes[4,38].

PG was only observed when the SP6 RNAP was co-encapsulated (Fig. 3d, e), demonstrating that genetic regulation of phospholipid synthesis occurs inside liposomes. In accordance with LUV experiments, activation of PG synthesis does not substantially decrease the amount of synthesized PE. In all cases, the final yield of PG was about two times lower than that of PE, mirroring the initial PE/PG ratio of the vesicle membranes. This result suggests that the homeostatic mechanism mediated by PssA takes place when lipid synthesis is compartmentalised inside liposomes. Moreover, we found evidence for accumulation of the phospholipid intermediates LPA, PA, and CDP-DAG, but not of PGP and PS (Supplementary Fig. 10).

We then aimed to expand the repertoire of synthesized phospholipids by mixing the 16:0 and 18:1 acyl-CoA precursors in equimolar amounts. We found that $82.9 \pm 0.4\%$ (without SP6 RNAP) and $79 \pm 11\%$ (with SP6 RNAP) of the total synthesized phospholipid end products contained mixed-chain products (PO) (Fig. 3d, Supplementary Fig. 10), which is significantly higher than the expected 50% assuming random chain incorporation. The fraction of synthesized dioleoyl ($8.8\% \pm 0.4\%$ without SP6 RNAP, $11\% \pm 9\%$ with SP6 RNAP) and dipalmitoyl ($8.2\% \pm 0.3\%$ without SP6 RNAP, $9\% \pm 5\%$ with SP6 RNAP) species was consequently low but appreciable. Concluding, it has been possible to selectively produce up to six different lipid species (DOPE, DOPG, DPPE, DPPG, POPE, POPG) with a one-pot reaction coupling gene expression and phospholipid synthesis within cell-sized liposomes.

**Visualization of membrane synthesis in individual liposomes.** In-liposome gene expression is subjected to high heterogeneity even when a single protein is produced from a high copy number of encapsulated DNA molecules[38]. While LC-MS methods provide sensitive detection of multiple lipid species in a liposome population, information about lipid composition at the single vesicle level is lost due to vesicle solubilisation. To overcome this limitation and to quantify the fraction of phospholipid-producing liposomes as well as the degree of heterogeneity, we established two fluorescence-based imaging assays. Moreover, optical microscopy methods gave us the opportunity to confirm our assumption that synthesized lipids are incorporated into the liposome membrane.

The first approach was based on the use of the nitrobenzoxadiazole (NBD)-labelled palmitoyl-CoA as a fluorescent substrate for phospholipid synthesis (Fig. 4a). The integration of the NBD-labelled acyl chain into the different enzymatic products was analysed by high-performance liquid chromatography (HPLC) (Fig. 4b). Peak assignment was realized by monitoring chromatograms of samples when only parts of the enzymatic pathway were expressed in the presence of LUVs. New peaks appearing after addition of a gene coding for an enzyme downstream the pathway were assumed to correspond to the final reaction product. In this way, signatures for the NBD-labelled PA, PS, and PE could unambiguously be identified (Fig. 4b, c). Furthermore, NBD-labelled PA and PE were detected when pGEMM7 was expressed inside cell-sized liposomes (Fig. 4b). These results demonstrate the versatility of our platform to synthesize novel lipid species.

Next, we performed fluorescence microscopy experiments to image the membrane localisation of newly synthesized NBD-labelled phospholipid species from the interior of liposomes. We reasoned that two-acyl chain phospholipid products conjugated to NBD are more stably inserted in the bilayer than mono acyl species (NBD-palmitoyl-CoA and NBD-LPA) that have a faster exchange rate between the membrane and the bulk phase.

Therefore, a more intense NBD signal at the liposome membrane is expected upon successful lipid production. A mixture of palmitoyl-CoA and NBD-palmitoyl-CoA (9:1 molar ratio) was used as acyl-chain precursors. This ratio was chosen to minimize the chance of incorporating two NBD-labelled chains in one phospholipid, which might result in fluorophore quenching, whilst yielding a sufficiently high fraction of NBD-labelled phospholipids for imaging. After pGEMM7 expression, the liposomes were diluted to reduce the membrane signal coming from NBD-palmitoyl-CoA and NBD-LPA. Background signal resulting from the transient interaction of NBD-palmitoyl-CoA with the vesicles was assayed in control samples where proteinase K was supplemented both inside and outside liposomes to totally inhibit gene expression (Supplementary Fig. 11). NBD-enriched liposomes, i.e. liposomes that successfully converted NBD-palmitoyl-CoA into two-acyl compounds were analysed. Expression of pGEMM7 inside liposomes led to a higher NBD signal at the membrane (Supplementary Fig. 11) and to a higher percentage of NBD-enriched liposomes than in the control sample (Fig. 4d) demonstrating phospholipid biosynthesis at the single vesicle level. In addition, the moderate increase (~50%) of the fraction of NBD-enriched liposomes when omitting the proteinase K (Fig. 4d) might be explained by an enhancement of enzymatic activity in liposome-confined reactions, as suggested above for lipid production at the population level (Fig. 3d, e).

The second strategy to detect lipid synthesis and membrane incorporation relies on the C2-domain of lactadherin fused to eGFP (LactC2-eGFP) as a PS-specific fluorescent reporter[39] (Fig. 5a, Supplementary Figs. 12 and 15). At a concentration of 150 nM, LactC2-eGFP binds to PS-containing membranes, but not to membranes where PS was substituted by PG (Supplementary Fig. 15). PS is not an end-product of our reconstituted lipid synthesis pathway and is rapidly converted by Psd into PE (Supplementary Figs. 9 and 10). To enable accumulation of PS, the plasmid DNA pGEMM7 was linearized using EcoRI that cuts at a unique restriction site located in the *psd* gene (Supplementary Fig. 16). The only end-product of the pathway encoded by the resulting construct (named pGEMM7Δ*psd*) is PS, when the SP6 RNAP is not added. Using pGEMM7Δ*psd* as a template for in-liposome gene expression led to significant accumulation of PS, as detected by LC-MS (Supplementary Fig. 16). Some residual PE synthesis was also measured, most likely as the result of incomplete restriction of the *psd* gene (Supplementary Fig. 16). When LactC2-eGFP was added to the feeding solution to probe PS production in individual liposomes, a clear recruitment to the membrane of some liposomes was observed (Fig. 5b, c), indicating PS-enrichment. No significant membrane binding of LactC2-eGFP was observed when omitting either oleoyl-CoA or the pGEMM7 template (Fig. 5b, d), corroborating the high PS specificity. Automated image analysis allowed us to extract the average rim intensity of eGFP in a large number of liposomes. A wide distribution of eGFP intensity values in PS-synthesizing liposomes was measured (Fig. 5d). The coefficient of variation is ~2-fold higher than in control samples with a predetermined fraction of PS (Supplementary Fig. 17). This result further supports the highly heterogeneous nature of liposome-encapsulated lipid synthesis. Moreover, we found that ~50% of the liposomes exhibited PS enrichment (Fig. 5e). Similar results were obtained when LactC2 was fused to mCherry in place of eGFP (Supplementary Note 3, Supplementary Figs. 13, 14, 17 and 18). We noticed that this approach is more robust and provides higher signal-to-background ratio than the use of an NBD-labelled acyl precursor. Moreover, no washing steps are necessary, making LactC2-eGFP a superior lipid probe to obtain kinetic information by real-time fluorescence imaging of individual liposomes. Figure 6a shows a representative liposome imaged at

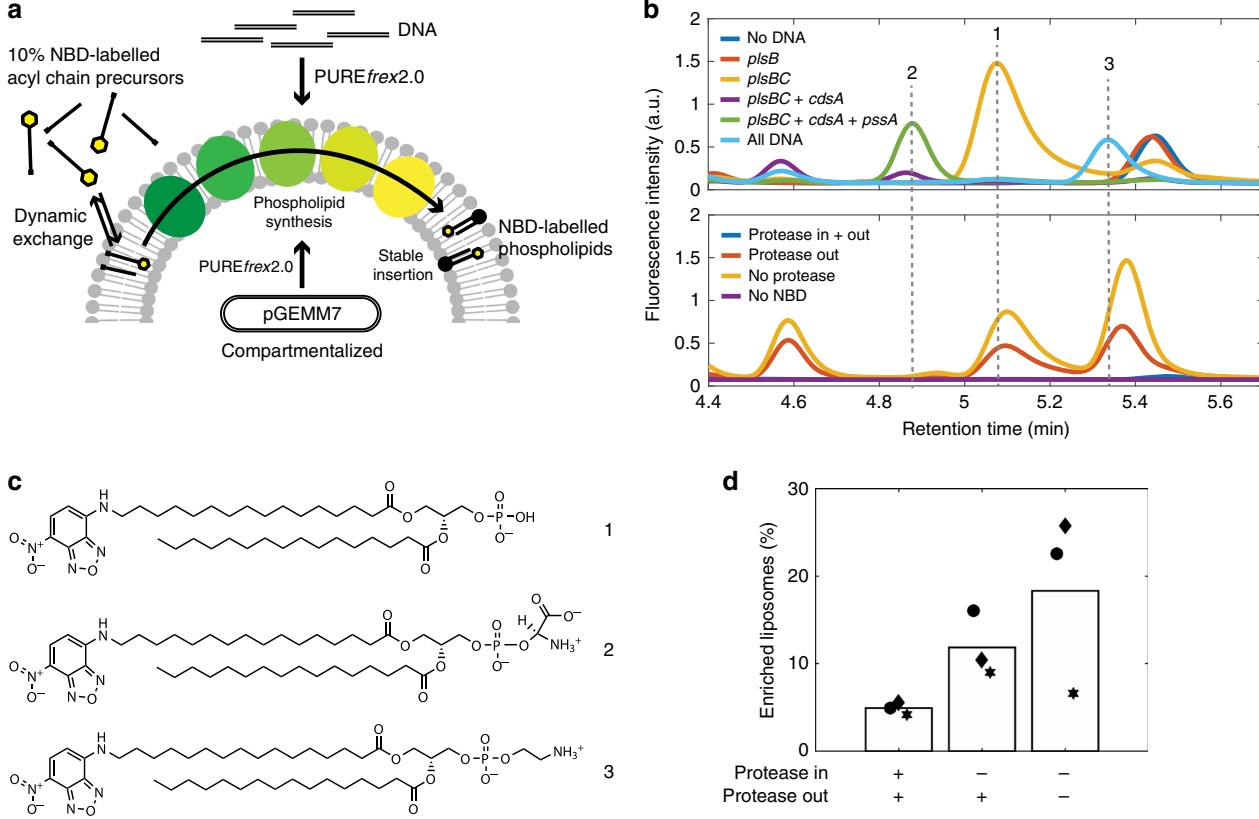

**Fig. 4 Cell-free biosynthesis of NBD-labelled phospholipids. a** Schematic illustration of the conversion of NBD-labelled acyl chain precursors into newly synthesized membrane-bound phospholipids. NBD-labelled acyl-CoA molecules undergo dynamic exchange between the membrane and the aqueous solution. Upon processing of the lipid precursors by the five-enzyme pathway, membrane-stable NBD-labelled PE is formed and an increase in NBD fluorescence at the liposome membrane is expected. Either PCR fragments of the individual genes or pGEMM7 were used as DNA templates and no SP6 RNAP was introduced. The precursor mix consisted of 10% (in mol) NBD-labelled palmitoyl-CoA and 90% palmitoyl-CoA. Cell-free gene expression was either performed outside LUVs or it was compartmentalised inside giant vesicles, as indicated. **b** HPLC chromatograms of NBD-conjugated species detected when single and multiple genes of the pathway are expressed in the presence of LUVs (top), and when pGEMM7 is expressed inside giant vesicles (bottom). The different gene combinations allowed us to assign peaks to specific lipid end products. Clear peaks were found for NBD-labelled DPPA (peak 1, *plsB* and *plsC*), NBD-labelled DPPS (peak 2, *plsB, plsC, cdsA*, and *pssA*), and NBD-labelled DPPE (peak 3, *plsB, plsC, cdsA, pssA*, and *psd*, labelled 'all DNA'). In the giant vesicle assay (bottom), proteinase K was either added to the inside of liposomes (no lipid synthesis), to the outside of liposomes (lipid synthesis restricted to the liposome lumen), or it was omitted. NBD-labelled DPPA and NBD-labelled DPPE could be observed. **c** Chemical structures of NBD-labelled DPPA (1), NBD-labelled DPPS (2), and NBD-labelled DPPE (3). **d** Percentage of NBD-enriched liposomes, i.e. liposomes that successfully converted NBD-palmitoyl-CoA into two-acyl compounds, was calculated by analysing line profiles of single liposomes imaged by fluorescence confocal microscopy (Supplementary Fig. 11). Conditions correspond to the giant vesicle experiment shown in (**b**) (bottom). The samples were washed three times to remove non-reacted NBD-palmitoyl-CoA. Bars are mean values from three independent experiments. Symbols indicate data points from individual repeats. A total of 741, 613, and 505 line profiles were analysed (from left to right). Source data are available for (**d**).

six time points. Between 0.5 and 6 h, a clear increase in LactC2-eGFP signal at the membrane can be observed. Plotting fluorescence intensity over time for 47 liposomes from three independent experiments shows a sigmoidal profile representing synthesis and membrane incorporation of PS, with a plateau time of ~4.5 ± 2.5 h and a rate of 9.2 ± 6.9 a.u. per minute (Fig. 6b, Supplementary Note 4, Supplementary Fig. 19). No clear dependency of the kinetic parameters with respect to the liposome size was observed for vesicles with an apparent diameter ranging between 4 and 12 μm (Fig. 6b, Supplementary Fig. 19). In addition, the amount of de novo synthesized lipids incorporated in the membrane was not sufficient for directly observing liposome growth under an optical microscope (Fig. 6c). When oleoyl-CoA was omitted, no increase of the LactC2-eGFP signal intensity was observed, confirming the specificity for synthesized PS. Further investigations will be necessary to elucidate the rate-limiting step of the LactC2-eGFP signal increase and the cause of saturation. In particular, it would be insightful to examine if

LactC2-eGFP recruitment saturates due to cessation of PS production.

## Discussion

We demonstrated here that an entire bacterial phospholipid synthesis pathway can be reconstituted inside liposomes by expressing seven membrane-associated enzymes from their genes concatenated on a DNA minigenome. Because the internally synthesized PE and PG lipids are also constituents of the liposome membrane, our synthetic cell platform satisfies the key requirements for self-maintenance. Moreover, higher-level regulation of membrane composition was provided through genetic control and metabolic feedback mechanisms, two processes that have so far been considered to be exclusive attributes of living organisms. The average PE-to-PG ratio could be maintained within the liposome population during phospholipid production, which is important to achieve homeostatic membrane growth.

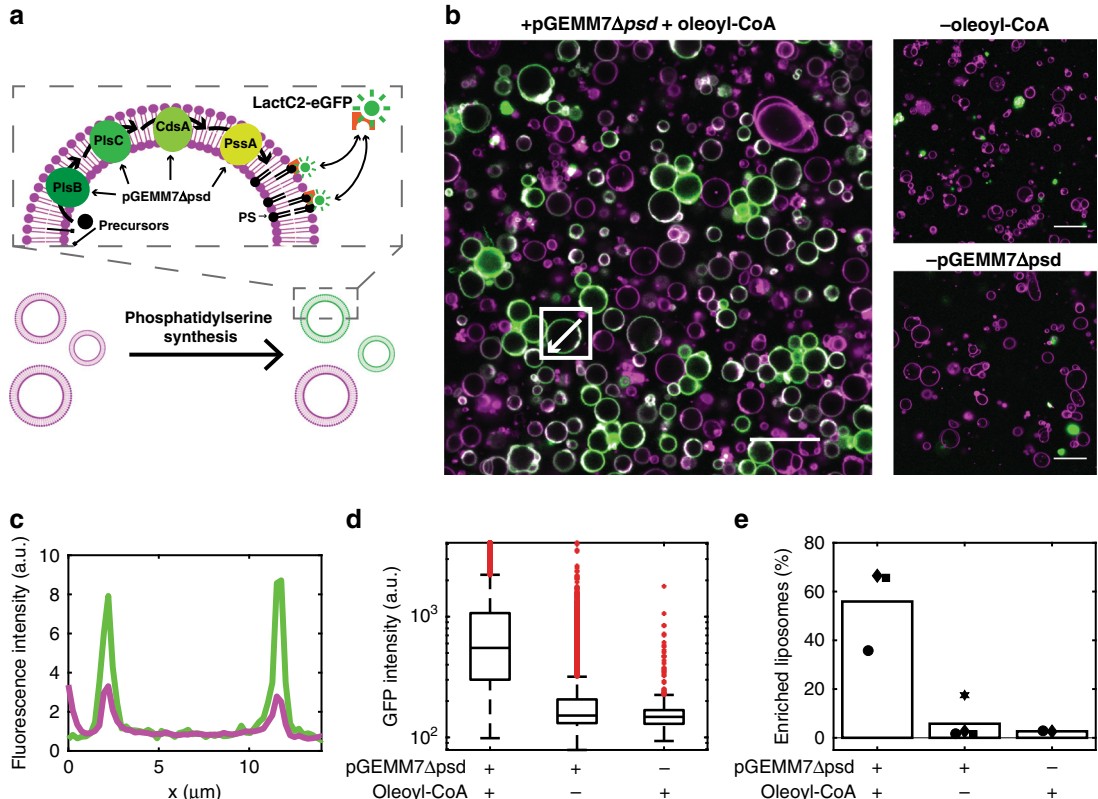

**Fig. 5 Single-vesicle imaging of internal PS production using LactC2-eGFP. a** Schematic representation of gene expression-coupled PS biosynthesis inside liposomes and fluorescence imaging using the PS-specific LactC2-eGFP probe. The linear pGEMM7Δ*psd* template was expressed within giant vesicles to produce the PlsB, PlsC, CdsA and PssA enzymes which catalyse formation of PS from acyl-CoA and G3P. Membrane-exposed PS recruits the fluorescent reporter LactC2-eGFP, resulting in accumulated GFP signal in PS-enriched liposomes. **b** Fluorescence confocal images of liposomes (membrane dye in magenta) producing DOPS from oleoyl-CoA, as illustrated in (**a**). The externally added LactC2-eGFP binds to PS-containing liposomes and stains the membrane in green, as observed in three independent repeats. In a series of negative control experiments, oleoyl-CoA was omitted (four independent repeats), or the pGEMM7Δ*psd* DNA was replaced by a DNA coding for an unrelated protein, namely the terminal protein of the ϕ29 phage (two independent repeats)[55]. Bright spots of clustered LactC2-eGFP molecules that do not co-localise with liposomes are sometimes visible. The LactC2-mCherry variant showed less propensity to form clusters than the eGFP fusion protein (Supplementary Fig. 18) and similar quantitative results were obtained with the two reporters (Supplementary Figs. 17, 18). Scale bars indicate 20 µm. **c** Line profiles of LactC2-eGFP intensity (green) and Texas Red membrane dye intensity (magenta) of the liposome highlighted in (**b**). **d** Box-plot representation of the single-vesicle average LactC2-eGFP intensity values for the indicated samples. Data were pooled from (from left to right) three, four and two independent repeats, corresponding to 4048, 3642 and 569 liposomes analysed, respectively. Membrane-localised GFP fluorescent intensity is significantly higher when both DNA and oleoyl-CoA were present (left) compared to negative controls without oleoyl-CoA (middle) or DNA (right), with $p < 0.0001$ (two-sample Welch's $t$-test). **e** Percentage of PS-enriched liposomes for the three types of samples analysed in (**d**) with results from independent experiments indicated by black symbols, with identical symbols referring to experiments performed in parallel. Bar height represents the mean percentage of PS-enriched liposomes. The amount of PS-enriched liposomes is significantly higher when both DNA and oleoyl-CoA were present compared to the two negative controls, with $p < 0.04$ (two-sample Welch's $t$-test). Source data are available for (**d**, **e**).

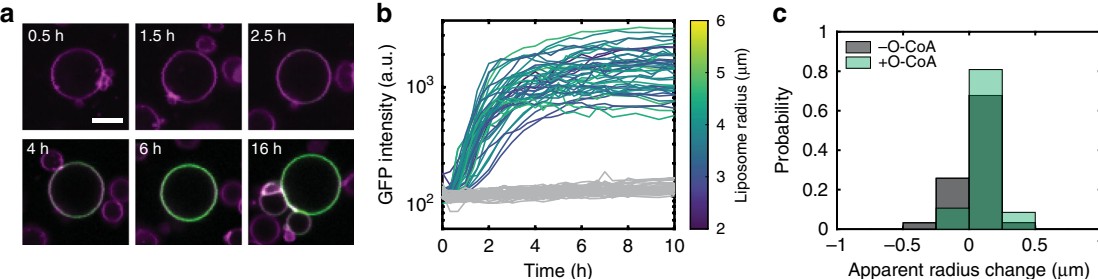

**Fig. 6 Kinetics of PS production and probe recruitment. a** Time-lapse images of a liposome exhibiting increasing LactC2-eGFP signal over time, as observed for 47 liposomes in three independent repeats. Scale bar is 5 µm; picture size is identical for all images. **b** Single-liposome kinetics of LactC2-eGFP binding for 47 PS-synthesizing liposomes, colour-coded with respect to the liposome radius (aggregated data from three independent repeats), and for 28 liposomes in a control sample that does not contain oleoyl-CoA (grey curves). **c** Probability distributions of the apparent radius change for the liposomes analysed in (**b**). The apparent radius change was determined by calculating the difference between the apparent radius at 0 h and 16 h. Identical imaging settings were used for all acquired data. Source data are available for (**b**, **c**).

What is the distribution of internally produced lipids in the bilayer? Phosphatidylserine is likely synthesized on the inner leaflet of the liposome membrane[40]. Nevertheless, synthesized PS is detected on the outer leaflet, where it is exposed to the LactC2-eGFP probe. Flipping of phospholipids is not energetically favourable and requires the assistance of specialized enzymes in vivo. However, the artificial bilayer of our liposomes is not as rigid as the bacterial cell membrane and is more prone to transient defects[38,41]. Therefore, membrane dynamic processes, such as lipid flip-flop and translocation of small molecules, may be less impaired in liposomes, facilitating partitioning of PS (and possibly of other synthesized lipids too) in the outer leaflet. Although the possibility that LactC2-GFP permeates across the membrane and binds PS exposed to the lumen cannot be excluded, this process is severely hindered by the bulky fusion protein.

To realize a full proliferation cycle, all membrane constituents should be co-synthesized. Here, we showed internal production of DOPE and DOPG, two out of the four membrane building blocks. Synthesis of the two other lipids, namely PC and CL, would require the reconstitution of only two additional proteins, PmtA and ClsA, respectively[42,43].

Besides having clear implications to creating a synthetic minimal cell, we envisage that our engineered liposomes could serve as a versatile platform for tailored biosynthesis of natural and artificial phospholipids of industrial or biotechnological value, such as lipids with asymmetric acyl chain compositions and acyl-labelled phospholipid analogues (Supplementary Table 7).

Single vesicle imaging revealed that a significant fraction of the liposomes does not display lipid enrichment (Figs. 4d, 5b, e). Moreover, a large heterogeneity in the level of synthesized phospholipids was observed among liposomes (Fig. 5d, Supplementary Fig. 17). Such a compositional and functional heterogeneity within a population of PURE system-containing liposomes has been reported in other studies[38,41,44,45] and is inherent to gene expression in cell-sized compartments. In the present experiments, other sources of heterogeneity in lipid enrichment may also contribute, such as a variability in the adsorption of acyl-CoA among liposomes upon resuspension of the precursor film. Investigating the mechanisms leading to phenotypic differences will be important to further optimise the chain of reactions from genes to output lipids.

Synthesis of phospholipids from an internal machinery and their incorporation in the lipid bilayer are essential steps toward physical growth. However, no visible membrane or volume expansion could unambiguously be measured by optical microscopy. It is clear that the amount of synthesized lipids per liposome should be increased to achieve doubling of the membrane surface area, which is necessary for sustainable proliferation. We envisage two complementary strategies to overcome this limitation, one acting at the gene expression level, the other at the lipid biosynthesis level. First, one could strive to improve the PURE system performance for producing larger amounts of the encoded enzymes in liposomes. Given the limited knowledge about the biochemical steps governing PURE system reactions[46], it remains nevertheless challenging to find generic solutions for improved DNA sequence design and composition of the PURE system. Alternatively, employing the native E. coli RNA polymerase along with sigma factors would expand the capabilities to regulate the transcription of individual genes and fine-tune the level of individual proteins[47,48]. This could, in turn, ameliorate the production rate and yield of the output lipids.

Another factor that might limit the final amount of synthesized phospholipids is the initial concentration of acyl-CoA, absolute and relative with respect to the concentration of liposomes. Adding more than 100 μM acyl-CoA is hardly feasible due to its poor solubility in the presence of high concentration of $Mg^{2+}$ contained in the PURE system and to its detergent effect on liposome membranes. One solution would be to provide a continuous supply of low-concentration acyl-CoA. Alternatively, more soluble precursors, such as acyl-ACP, fatty acids and malonyl-CoA could be used. Expanding the pathway upstream by introducing the FadD enzyme would enable to substitute acyl-CoA with a fatty acid and CoA[21]. Finally, the highly soluble malonyl-CoA could be used as a substrate provided the eukaryotic FASII mega-protein can be synthesized in a functional state in the PURE system[49].

Alternatively, chemical synthesis of non-natural phospholipids has emerged as an interesting strategy because of its high yield and quick conversion[12–14]. Chemical synthesis could potentially be coupled to one or more enzymatic reactions, resulting in a hybrid system equipped with genetic control facilitating rapid lipid synthesis[50]. A radically different approach would consist to use alternative membrane constituents, such as amphiphilic peptides, that would be expressed from the inside of the vesicle[51].

DNA-programmed lipid synthesis could be exploited as a rudimentary mechanism to trigger division of liposomes. Budding events reminiscent of the proliferation mode of L-form bacteria[52] could be stimulated through an excess membrane synthesis, potentially aided by gentle shear forces. Furthermore, internal synthesis of membrane remodelling phospholipids like DLPE, produced when starting from the short-chain 12:0 acyl-CoA, in combination with temperature cycling[53], might assist vesicle deformation and division.

It has not escaped our attention that liposome-confined DNA-based phospholipid production, combined with the fluorescently tagged LactC2 as a selection marker, is amenable to directed evolution experiments, owing to the linkage between genotype and phenotype. Activity of single or multiple enzymes in the pathway, or substrate selectivity, could be improved by generating a library of mutagenized genes and selecting for PS-enriched liposomes by fluorescence-activated cell sorting[54]. This strategy may become decisive when combining membrane growth with other functional modules, such as DNA replication[55] and liposome division[56,57].

## Methods

**Buffers and solutions**. All buffers and solutions were made using Milli-Q grade water with 18.2 MΩ resistivity (Millipore, USA). Chemicals were purchased from Sigma-Aldrich unless otherwise indicated.

**Design and assembly of the pGEMM7 plasmid**. The plasmid pGEMM7 was assembled from seven PCR fragments containing independent transcriptional cassettes and the plasmid backbone of pUC19 (New England Biolabs, USA) (Supplementary Table 1). The genes were used in a previous study with each gene inserted in a separate DNA construct[23]. Individual genes were amplified by PCR using primers containing linker sequences to determine the order and orientation of each cassette in the final plasmid. Linker sequences of 30 bp were designed by a random DNA generator such that they had no or little homologies to the E. coli genome (R20DNA designer, https://www.syntegron.org/R20/R20/R20.html, Imperial College London) to minimize unwanted recombination events. The vector backbone was amplified using primers 829 and 830 giving a 1932 bp product bearing either linker site 1 or 13 on the ends. The transcriptional cassette of plsB was amplified using primers 628 and 629 introducing linker site 1 upstream of the gene and linker site 2 downstream. All other remaining transcriptional cassettes were made in the same way adding linker sequences both upstream and downstream of the cassette to enable each cassette to be recombined with the next one by Gibson assembly[31]. Primer 819 also adds an SP6 promoter to the pgsA gene as well as a linker sequence. The second SP6 promoter sequence was added to the pgpA gene in a previous step using primer 817. The homologous site that was added using primer 817 was deleted in a subsequent PCR using primer 851. Supplementary Fig. 1 shows a schematic drawing of the two-step process to incorporate the homologous linker sites by PCR and then using the individual fragments to assemble pGEMM7. The primers, their targets and the homologous site they are bearing are listed in Supplementary Table 2.

**Cloning of the pGEMM7 plasmid.** All fragments for Gibson assembly were amplified using Phusion High-Fidelity DNA polymerase (New England Biolabs, USA) with the recommended standard reaction conditions from the supplier. Elongation times and primer annealing temperatures were varied according to primer length between 55 °C and 65 °C. Primers and remnants of the PCR reaction were removed using the Wizard PCR cleanup kit (Promega, USA). The concentration of the purified DNA was determined using an ND-2000 NanoDrop spectrophotometer. Purified PCR products were mixed following the pipetting scheme in Supplementary Table 3 plus 15 μL of prepared Gibson assembly mix containing 100 mM Tris-HCl, 50 mM MgCl$_2$, 0.2 mM each dNTP, 10 mM dithiothreitol (DTT), 5% w/v PEG-8000, 1 mM nicotinamide adenine dinucleotide (NAD), 5.33 U mL$^{-1}$ T5 Exonuclease, 33.3 U mL$^{-1}$ Phusion polymerase and 5.33 U mL$^{-1}$ Taq-ligase in a final volume of 20 μL. The Gibson assembly mixture was incubated at 50 °C for 1 h and 5 μL were subsequently used for transformation of 50 μL One Shot™ TOP10 Chemically Competent E. coli cells (ThermoFisher Scientific, USA, catalogue number C4040-10).

Transformed cells were recovered in 1 mL LB medium for 1 h and transferred on LB-Agar plates containing 50 μg mL$^{-1}$ ampicillin. After overnight incubation at 37 °C, ten colonies were selected for colony PCR using primers 91 and 397 which bind in the T7 terminator region and the RBS, respectively. Four of the tested colonies gave the expected pattern (Supplementary Fig. 3a) and were subsequently grown overnight in LB medium. Their plasmid DNA was isolated using a PureYield miniprep kit (Promega, USA) and was further analysed with restriction digestion using the enzymes EcoRI-HF, SacI and DraI (New England Biolabs, USA). Supplementary Fig. 3b shows that all four colonies gave the expected pattern consisting of digestion products of 4300 bp, 2836 bp, 1863 bp, 1395 bp, 692 bp, and 19 bp (indicated by black stars, only the 19-bp product was not visible), plus some side products attributed to incomplete DNA digestion. The correct DNA sequence was finally confirmed with Sanger sequencing (Macrogen, South-Korea).

**Cloning of eGFP-lactC2 and plasmid purification.** The original plasmid containing the egfp-lactC2 gene was described in ref. [39] and was kindly provided by the lab of Dorus Gadella (University of Amsterdam, Netherlands). To enable expression and isolation from E. coli, regular PCR reactions were performed to amplify both the plasmid backbone of a pET11a vector and the egfp-lactC2 gene construct. Primers 471 (forward) and 850 (reverse) were used for the amplification of the pET-11a backbone. Primers 848 (forward) and 849 (reverse) were used for the amplification of egfp-lactC2. The reaction was performed with 10 ng of template DNA, 1 U of Phusion High-Fidelity DNA Polymerase (New England Biolabs) in HF buffer and supplemented with 0.2 mM of dNTPs, and 0.2 μM of both forward and reverse primers in a final volume of 50 μL. An initial heating step at 95 °C for 5 min was applied to allow denaturation of DNA. The PCR reaction consisted of 34 cycles of 30 s steps for melting DNA at 95 °C, followed by the hybridization of the primers for 30 s at 55 °C and the elongation by the DNA polymerase at 72 °C for 30 s per kb template. After the 34 cycles, the temperature was kept at 72 °C for 5 min. Both PCR products were purified using the Wizard PCR cleanup kit (Promega, USA).

The size of the PCR products was verified on an TAE agarose gel (1% w/v) using SYBR safe staining (Thermo Fisher). The BenchTop 1-kb DNA Ladder from Promega was used. The fragments corresponding to the adequate sequence lengths of 1.3 kb and 5.6 kb were excised from the gel and purified using the Promega Wizard SV Gel and PCR Clean-Up System kit. DNA concentration of the eluate was determined by measuring the absorbance at 260 nm with a NanoDrop 2000c.

The pET-11a backbone and egfp-lactC2 gene fragments were assembled using Gibson assembly[31]. 100 ng of backbone and an equimolar amount of the egfp-lactC2 PCR fragment were mixed in a solution containing 100 mM Tris-HCl, 50 mM MgCl$_2$, 0.2 mM each dNTP, 10 mM DTT, 5% w/v PEG-8000, 1 mM NAD, 5.33 U mL$^{-1}$ T5 Exonuclease, 33.3 U mL$^{-1}$ Phusion polymerase and 5.33 U mL$^{-1}$ Taq-ligase in a final volume of 20 μL. The assembly reaction was incubated at 50 °C for 60 min. Then, 20 U μL$^{-1}$ of DpnI restriction enzyme (New England Biolabs, USA) were added to digest possible methylated DNA left and the mixture was incubated for an additional 15 min at 37 °C.

Five microliters of the assembly mixture were transformed into 50 μL of One Shot™ TOP10 chemically competent E. coli cells using heat shock. The cells were heat shocked in a water bath at 42 °C for 45 s and then transferred back to ice for 2 min, to reduce cell damage. After incubation in 1 mL of LB medium (1:20 dilution) for 20 min at 37 °C, 50 μL of the cell suspension were spread in LB plates supplemented with 50 μg mL$^{-1}$ ampicillin. The remaining sample was pelleted, resuspended in 50 μL of LB medium and plated. All plates were incubated overnight at 37 °C.

Six colonies were picked to perform colony PCR and a replica plate was made. A PCR reaction was performed with 0.5 U of GoTaq DNA Polymerase in GoTaq Buffer (both from Promega) supplemented with primers and dNTPs to a final volume of 20 μL. Adequate forward and reverse primers (25 and 310, respectively) were chosen to amplify the gene region and part of the backbone sequence upstream and downstream of the gene (Supplementary Table 2). DNA was purified using the Promega Wizard® SV Gel and PCR Clean-Up System and analysed on gel. Colonies leading to a band with the predicted length (6.9 kb) were grown in 5 mL LB medium overnight and plasmid DNA was isolated using the PureYield Plasmid Miniprep System (Promega). The plasmids were further tested by a restriction enzyme digestion analysis, in which 2.5 U of DraI and 2.5 U of StuI were mixed with 500 ng of DNA, in a final volume of 20 μL (both enzymes were from New England Biolabs). The mixture was then incubated at 37 °C for 1 h. Digested DNA was separated in TAE agarose gel (1%).

To infer the quality of the construct on the sequence level, DNA extracted from the six colonies was sequenced by Sanger sequencing (Macrogen). To 300 ng of plasmid DNA, 0.25 μM of adequate primers (288 and 25, Supplementary Table 2) were added, in a final volume of 10 μL. Plasmids with the correct sequence were selected.

**Overexpression and purification of LactC2-eGFP and -mCherry.** E. coli Rosetta ER2566 cells (New England Biolabs) and Rosetta 2 cells (Novagen) suited for protein overexpression were transformed with the plasmid for LactC2-eGFP by heat shock. The plasmid for LactC2-mCherry was transformed into Rosetta 2 cells and isolated in the same way as described below. A preculture of these strains was incubated overnight at 37 °C in LB medium supplemented with 50 μg L$^{-1}$ ampicillin. Then, the cultures were diluted in the same medium in a ratio of 1:1000 and incubated at 37 °C with agitation (200 rpm) until an OD$_{600}$ of ~0.6 was reached. Protein production was induced with 1 mM isopropyl β-D-1-thiogalactopyranoside. The cells were incubated at 30 °C for 3 h under agitation (200 rpm) and were pelleted by centrifugation at 16,000 × g for 5 min. The pellet was resuspended in buffer A (150 mM NaCl, 20 mM imidazole, 20 mM Tris pH, 7.5) and the cells were disrupted by sonication using ten pulses of 10 s and 30 s of interval, with 30% amplitude. After centrifugation at 4 °C for 15 min and 16,000 × g, the supernatant was cleared from debris.

Protein purification was done using Ni-NTA Spin Columns (Qiagen) following the supplier recommendations. The column was equilibrated and washed with buffer A and the protein was eluted with buffer B (150 mM NaCl, 500 mM imidazole, 20 mM Tris, pH 7.5). The elution buffer was exchanged for the storage buffer (10 mM Hepes-KOH, pH 7.5) using Zeba Spin Desalting Columns (ThermoFischer). This size-exclusion chromatographic spin down columns retain small molecules (<1 kDa) and recover mostly large molecules (>7 kDa). Throughout all the steps of protein purification and buffer exchange, samples were harvested for subsequent analysis in polyacrylamide gels.

The 12% polyacrylamide resolving gel and the 4% stacking gel were prepared with final concentrations of 0.12% of sodium dodecylsulfate, 150 mM of Tris-HCl, pH 8.8 for the resolving gel and 10 mM of Tris-HCl, pH 6.8 for the stacking gel. Ammonium persulfate and tetramethylethylenediamine were added after to begin polymerisation. The loading solution consisted of 15 μL of the protein sample mixed with 1 μL DTT and 15 μL Laemmli 2× Concentrate Loading Buffer (Sigma-Aldrich), and denatured at 95 °C for 10 min. The gel was run first at 100 V for 15 min and then at 180 V for ~45 min. Running buffer consisted of 250 mM Tris-HCl, 200 mM glycine, 1% w/v SDS, pH 8.3.

The concentration of the protein was measured with a Bradford assay. Bovine serum albumin was used as a standard spanning seven concentrations from 0.25 mg mL$^{-1}$ to 2 mg mL$^{-1}$. Each sample was assayed in triplicate, including a Milli-Q sample, and the absorbance at a wavelength of 595 nm was measured by spectrophotometry.

**Proteomics.** A targeted proteomics approach was used following established in-house protocols. Samples of PUREfrex2.0 (GeneFrontier, Japan) of 1 μL were incubated at 55 °C for 20 min in 16.5 μL of 50 mM Tris-HCl, 0.1% 2-octoglycoside, 12.5 mM DTT and 1 mM CaCl$_2$. Then, 32.6 mM final concentration of iodoacetamide was added and the solution was incubated for 30 min in the dark. Finally, 0.5 μg of trypsin was added and the solution was incubated overnight at 37 °C. The following day, 2 μL of 10% trifluoroacetic acid was added, the sample was incubated at room temperature for 5 min, the solution was centrifuged at 16,000 × g for 30 min and the supernatant was transferred to an HPLC-vial for analysis.

Mass spectrometry analysis of tryptic peptides was conducted on a 6460 Triple Quad LC-MS system using the MassHunter Workstation LC/MS Data Acquisition Software (Agilent Technologies, USA). From the samples prepared according to the protocol described above, 10 μL was injected into an ACQUITY UPLC® Peptide CSH™ C18 Column (Waters Corporation, USA). Peptides were separated in a gradient of buffer C (25 mM formic acid in Milli-Q) and buffer D (50 mM formic acid in acetonitrile) at a flow rate of 500 μL per minute at a column temperature of 40 °C. The column was equilibrated with 98:2 ratio of buffer C to D. After injection, over 20 min the ratio was changed to 75:25 buffer C to D after which, within 30 s, the ratio went to 20:80 buffer C to D and was held for another 30 s. Finally, the column was flushed for 5 min with 98:2 buffer C to D ratio. Supplementary Table 5 shows the transitions of the MS/MS measurements that were observed in every experiment. EF-Tu is a constant component of the PURE system and served as a global internal standard for variations due to evaporation or sample handling. All data were represented as the peak integrated intensity of a given peptide normalised to that of the TTLTAAITTVLAK peptide of EF-Tu. All proteomics results were analysed in Skyline-daily 4.1.1.18179 (MacCoss lab, University of Washington, USA).

Retention time was predicted after standard runs with the above-described method using the Pierce™ Peptide Retention Time Calibration Mixture (Catalogue number 88320, Thermo Scientific, USA).

**Precursor films.** Palmitoyl-CoA, oleoyl-CoA, and NBD-palmitoyl-CoA were obtained from Avanti Polar lipids (USA) in powdered form. The powders were dissolved in chloroform:methanol:water (40:10:1 vol. fractions), aliquoted, dried, and stored under argon. Before use, the acyl-CoA's were resuspended and diluted in chloroform to a final concentration of 100 μM. Using Gilson Microman pipettes, the acyl-CoA solution was added to PCR tubes. Organic solvent was evaporated at ambient pressure and temperature for ~5 h, resulting in a dried precursor film. Acyl-CoA volumes were chosen such that the concentration of precursor after resuspension in the samples was 100 μM (50 μM for NBD experiments). For NBD experiments, films consisted of 10% NBD-palmitoyl-CoA and 90% palmitoyl-CoA and were limitedly exposed to light.

**LUV experiments.** LUVs were prepared by extrusion of large multilamellar vesicles (LMVs). A 2 mg lipid mixture consisting of DOPC/DOPE/DOPG/CL/DSPE-PEG-biotin (50 mol%/36 mol%/12 mol%/2 mol%/1 mass%) dissolved in chloroform was prepared in a 2 mL glass vial, dried under gentle argon flow and subsequently desiccated for 1 h. The film was then resuspended in 250 μL buffer E (20 mM HEPES, 180 mM potassium glutamate, 14 mM magnesium acetate, pH 7.6) and vortexed to create LMVs. Four freeze-thaw cycles were applied and samples were extruded with a 400 nm membrane using the Avanti mini-extruder, according to instructions provided by the manufacturer. LUVs were aliquoted, snap-frozen in liquid nitrogen and stored at −20 °C.

PURE*frex*2.0 reaction solutions were assembled according to the instruction provided by the manufacturer, and supplied with 0.75 U μL⁻¹ Superase (Invitrogen), 5 mM β-mercaptoethanol, 500 μM ¹³C-labelled G3P, 1 mM CTP, 500 μM L-serine, 0.4 mg mL⁻¹ lipids from the LUV mixture, and 1 nM of pGEMM7 plasmid, unless stated otherwise. When indicated, 2 U μL⁻¹ SP6 RNAP was supplemented. The reaction mixture was then added to the dried precursor film and incubated overnight at 37 °C.

For the experiments shown in Fig. 2, LUVs with lipid compositions DOPC:DOPE (50:50) and DOPC:DOPG:CL (50:48:2) were prepared as described above, and were mixed in various ratios. Membrane fusion was promoted by applying four freeze-thaw cycles.

**In-liposome gene expression assays.** Giant vesicles were prepared according to the lipid-coated glass beads method[38]. 2 mg lipids consisting of DOPC/DOPE/DOPG/CL/DHPE-Texas Red/DSPE-PEG-biotin (50 mol%/36 mol%/12 mol%/2 mol%/0.5 mass%/1 mass%) were mixed with 25.4 μmol rhamnose in methanol and the mixture was added to 0.6 g of 212–300 μm glass beads (acid washed). Beads were rotary evaporated for 2 h at room temperature and 20 mbar, and desiccated overnight to remove the residual organic solvent. The lipid-coated beads were stored under argon at −20 °C up to one month.

PURE*frex*2.0 reaction solutions were assembled similarly to LUV experiments. Per 10 μL PURE*frex*2.0 reaction mixture, 5 mg (10 mg for LactC2-eGFP experiments) of lipid-coated beads were added. Lipid film swelling was performed for 2 h on ice with gentle tumbling every 30 min. Four freeze-thaw cycles were applied by dipping the sample in liquid nitrogen and thawing at room temperature. The supernatant (corresponding to about 50% of the total volume) was transferred to an Eppendorf tube using a cut pipette tip to avoid liposome breakage. To confine gene expression reactions to the inside of liposomes, 50 μg μL⁻¹ proteinase K was added to the liposome sample, unless indicated otherwise. For experiments involving LactC2-eGFP, 2 μL of liposome-containing supernatant were diluted in 5.5 μL of a feeding solution consisting of PURE*frex*2.0 Solution I and Milli-Q (3:7), 150 nM of LactC2-eGFP and 0.07 U μL⁻¹ RQ1 DNase (Promega). Liposomes were then transferred to the tube with deposited dried precursor films. Reactions were incubated overnight at 37 °C, or, in the case of time-lapse microscopy, liposomes were immediately immobilized for imaging (see below).

**Sample preparation for LC-MS and HPLC.** A solution consisting of methanol with 5 mM EDTA and 2 mM acetylacetone was prepared fresh for every experiment. Samples were diluted 10- (for HPLC) or 100-fold (for LC-MS) in the methanol solution, sonicated for 10 min, and centrifuged for 5 min at 16,000 × g. The supernatant containing the lipid fraction was transferred to Agilent 2 mL glass mass spectrometry vials with a low-volume inset, flushed with argon and stored at −20 °C. Samples were analysed within one week from preparation.

**LC-MS analysis of lipids.** Mass spectrometry measurements of phospholipid samples were performed using a 6460 Triple Quad LC-MS system equipped with a similar ACQUITY UPLC® Peptide CSH™ C18 Column as used in proteomics. However, different columns were used for each application, to prevent cross-contamination. Separation of lipids was performed using a gradient of mobile phase F (water with 0.05% ammonium hydroxide and 2 mM acetylacetone), and mobile phase G (80% 2-propanol, 20% acetonitrile, 0.05% ammonium hydroxide and 2 mM acetylacetone) at a flow rate of 300 μL min⁻¹ and a column temperature of 60 °C. To equilibrate the column, a ratio of mobile phase F to mobile phase G of 70:30 was used. Upon injection, this ratio was gradually changed to 100% mobile phase G over the course of 8 min and then kept like that for 2 min. Subsequently, over the course of 1 min, the initial 70:30 ratio of mobile phase F and G was reset, which was then used for the last 4 min of the run. The built-in autosampler of the

LC-MS system was used to inject 1 μL (quantitative analysis) or 5 μL (qualitative analysis of low-abundance compounds) of sample solution.

Transitions were established based on previous work[23], as well as scanning measurements of purified standards. The very regular fragmentation pattern (except for LPA and CDP-DAG fragmentation always occurs at the ester linkage between an acyl chain and the glycerol) could be used to determine transitions. Synthesized phospholipids were distinguished from phospholipids present at the start of the reaction as part of the liposome matrix by incorporation of ¹³C-G3P, resulting in a 3 Da (or 6 Da for PG) mass shift.

Mass spectrometry data were analysed using the Agilent MassHunter Workstation Software Quantitative Analysis program, which automatically integrates peaks corresponding to the transitions set in the method. Integrated peak intensities were exported to MATLAB R2016b (MathWorks) for further analysis. For each transition in each sample, the average integrated counts of two injections was determined. For end products, integrated counts were converted to concentrations using linear calibration curves fitted to signals from a dilution series of standards ran before and after every mass spectrometry measurement series.

**HPLC.** High-pressure liquid chromatography (HPLC) was used to separate synthetized NBD-labelled lipid species. An Agilent Technologies 1260 Infinity HPLC system equipped with an HSS T3 2.5 μm column was used with mobile phase H (60% acetonitrile, 40% water, 0.0114% formic acid, 7 mM ammonium formate, and 2 mM acetylacetone) and mobile phase I (90% 2-propanol, 10% acetonitrile, 0.0378% formic acid, and 2 mM acetylacetone), as previously reported in ref. [23]. The flow rate was 500 μL min⁻¹ and the column temperature was 35 °C. Upon injection of 5 μL of sample, 100% mobile phase H was used, over the course of 1.5 min changing to a ratio of mobile phase H to mobile phase I of 35:65, which was then gradually changed in 8.5 min to 30:70, and then, in 2 min, to 5:95, which was retained for 1 min. Subsequently, in the final 2 min of the run, the initial gradient was restored. NBD fluorescence was detected with an excitation wavelength of 463 nm and an emission wavelength of 536 nm.

**Microscopy.** Liposomes were immobilized in custom-made glass imaging chambers pre-incubated for 10 min with BSA-biotin:BSA (1 mg mL⁻¹) and then with Neutravidin (1 mg mL⁻¹). When appropriate, free NBD-palmitoyl-CoA was removed by washing the sample three times with an equal volume of buffer E, followed by 30 min of incubation at 37 °C. Image acquisition was performed using a Nikon A1R Laser scanning confocal microscope, operated via the NIS Elements software (Nikon), using the following excitation/emission wavelengths: 457/525 nm (NBD), 488/509 nm (LactC2-eGFP), 514/540 nm (YFP), and 561/595 nm (Texas Red). The sample height was adjusted manually in order to equatorially dissect as many liposomes as possible. Within data sets, identical imaging settings were always used.

**Image analysis.** To determine NBD and LactC2-eGFP fluorescence intensity at the membrane, both manual and automated image analyses have been applied concurrently. For manual image analysis, Fiji[58] was used to obtain line profiles of Texas Red and NBD/eGFP intensity along cross-sections of liposomes selected for unilamellarity in the membrane dye channel. To prevent bias, the NBD/eGFP channel was not viewed during analysis. The two peaks in the NBD/eGFP line profiles were subsequently detected using a custom MATLAB R2016b script, and the average intensity of these peaks was calculated. Line profiles with less or more than two peaks were discarded from the analysis.

The automated image analysis script was written in MATLAB R2016b and was based on the image analysis procedure we previously developed[38]. In short, a *floodfill.m* algorithm was used to determine liposome lumina, based on the Texas Red membrane signal. To determine the NBD/eGFP intensity along the membrane, first the centroid and radius were determined for every detected liposome. Then, intensity profiles along a line from the centroid to 1.5 times the radius, along 63 different angles, were determined. For every line profile, the maximum intensity, corresponding to the membrane intersection, was recorded, and values were averaged to obtain the NBD/eGFP intensity of the membrane. Since this approach is quite sensitive to possible deviations from a spherical shape, a more stringent circularity criterion than previously reported was applied.

For NBD fluorescence analysis (Fig. 4d), liposomes were selected for unilamellarity based on the Texas Red channel. NBD-enrichment was defined when the mean NBD peak fluorescence for a given liposome is higher than the mean plus two standard deviations of the signal in the 'proteinase K in' negative control.

In Fig. 5e, liposomes were defined as enriched in PS if their average LactC2-eGFP intensity is higher than the mean plus two standard deviations of the intensity distribution obtained with liposomes containing 0% PS (see Supplementary Fig. 15). In Fig. 6b, the apparent radius was calculated based on the perimeter of the liposome selection marker under the assumption of a perfect circle. A bin width of 250 nm was chosen.

**Statistics.** Box-plots in Fig. 5d have the following characteristics. The middle line is the median, and the bottom and top ends of the box represent the first and third quartiles respectively, with the length of the box indicating the interquartile range. Whiskers extend to the lowest point less than 1.5 interquartile range from below

the first quartile (lower box end) and to the highest point less than 1.5 interquartile range above the third quartile (upper box end). Outlying points with LactC2-eGFP intensity beyond the upper whisker are plotted individually as red dots.

**Reporting summary**. Further information on research design is available in the Nature Research Reporting Summary linked to this article.

## Data availability

Data supporting the findings of this paper are available from the corresponding author upon reasonable request. A reporting summary for this Article is available as a Supplementary Information file. Proteomics data are uploaded on Panorama Public (https://panoramaweb.org/) and are available using access URL (https://panoramaweb.org/Q4XGTX.url) or ProteomeXchange ID PXD020399. Source data are provided with this paper.

## Code availability

The custom MATLAB image analysis script used to extract fluorescence intensity at the liposome membrane can be found at: https://doi.org/10.5281/zenodo.3923781.

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

## Acknowledgements

We thank Andrew Scott for performing preliminary experiments with NBD-palmitoyl-CoA, Marijn van den Brink for contributing experiments with LactC2-mCherry, Jorick van de Grift from the Dorus Gadella lab (University of Amsterdam) for providing us with the original plasmid containing the *egfp-lactC2* gene, Niels van den Broek for assistance with LC-MS, Gemma van der Voort for contributing earlier versions of the pGEMM constructs, and Sophie van der Horst for contributing experiments for the promoter orthogonality assay. This project was funded by the Netherlands Organization for Scientific Research (NWO/OCW) through the Gravitation grants 'NanoFront—Frontiers of Nanoscience' and 'BaSyC—Building a Synthetic Cell' (024.003.019).

## Author contributions

C.D. conceived and supervised the research. D.B., D.F. and A.C.S. performed the experiments. D.B., D.F. and C.D. designed the experiments and wrote the paper. All the authors analysed data and discussed the results.

## Competing interests

The authors declare no competing interests.
