## [Peer Review File · Nature Communications]

Reviewers' comments:

Reviewer #1 (Remarks to the Author):

In this manuscript the Danelon laboratory does a good job of developing a cell-free system for the synthesis of lipids. Such work is of clear importance to the synthesis of synthetic cells and may have additional biotechnological applications. The experiments are generally well designed and the data are convincing.

This group did publish previously in PLOS ONE similar work that demonstrated the synthesis of lipid. It is not immediately clear what the difference is with this submitted manuscript. The authors should do a better job in indicating what the new advance is.

I think this work is important and will be used by several laboratories as they try to build artificial cells. I do, however, have suggestions that the authors may wish to consider during revisions.

1. I do not believe that the authors ever demonstrated that the membrane proteins that were cell-free expressed actually partitioned to the membrane. Functional activity is certainly important, but it would be nice to demonstrate the localization of these proteins since that may aid future work.
2. Liposome growth was not demonstrated but could have easily been observed with FRET labeled lipids. This would strengthen the manuscript considerably since the goal appears to be growth.
3. The figures frequently lack statistical analysis and an indication of the number of replicates. Sometimes the units of the axes are missing.
4. The first two sentences of the abstract are probably a little too strong regarding what is needed for life.
5. The introduction begins by stating that life is an individual construct, but that is again, likely too strong. Evolution gives rise to populations. I know what the authors mean, but more care should be taken in how these ideas are expressed.
6. Last sentence of the introduction: "Our results provide experimental evidence for DNA-encoded homeostatic growth of a liposome-based artificial cell." But the authors do not demonstrate growth in this manuscript.
7. It is not clear why data was split between figure 2 and Supplementary Figure 11. The two should be combined together.
8. The degradation of material external to the liposomes was either with proteinase K or DNase I. Is there a reason why both weren't used together?
9. Line 5, page 10: It is unclear what is meant by "hidden in ensemble measurements."
10. Supplementary Figure 4: Why are the lipid concentrations in arbitrary units?
11. Supplementary Figure 4: The legend states "We hypothesized that this was caused by transcription termination read-through by the T7 RNAP, transcribing downstream PG synthesis genes even if those are under control of an orthogonal promoter." One way to avoid this problem is to clone the different modules in different directions.
12. Supplementary Figure 8: without error bars, the data are hard to interpret
13. Figure 3b: The difference between the dashed lines is confusing and not enough to appreciate the logic of the experiment.

Reviewer #2 (Remarks to the Author):

In this study, Blanken and colleagues constructed a single plasmid encoding seven genes involved in lipid synthesis and showed protein synthesis using a reconstituted transcription/translation system (PURE system) from the information. The results showed that the 7 protein involved in lipid synthesis was successfully synthesized in the liposomes, and indicated the synthesis of the objective lipids. The direction in which this research aims is certainly important in the future of synthetic biology. However, questions remain regarding the achievement, novelty of the thesis, and conceptual advances.

The authors conclude in the title "membrane synthesis", but no evidence has been presented beyond lipid synthesis as the author mentioned that they failed to observe an increase of liposomes sizes by their system. This is in contrast to reference 19 that shows this point. Regarding novelty, previous studies have already developed technologies to combine seven or more genes into one plasmid. Expression and insertion to the membrane of the 7 proteins involved in lipid synthesis have been shown in their previous study. Although the only novel point of the manuscript is the expression of the 7 proteins in liposomes, there are many reports that showed normal protein synthesis (including membrane proteins as the author already reported) via the PURE system within liposomes. Due to the levels of achievement they set and lack of significant novelty and conceptual advances, I do not support the current manuscript for publishing in Nature Communication.

Other major comments

L2, p3. Figure 5 shows synthesis of PS within GUVs. However, PS is also not part of the original membrane composition of *E. coli* system.

L9, p3. No evidence of the growth of liposome is shown in this manuscript.

L27, p5. I think normalization using a different peptide fragment is a non-standard method in MS-based quantification. Do the authors have a reference or data that support the feasibility of the normalization?

Figure 3&4. I wonder why the protease or DNase outside liposomes affect protein expression within liposomes. Does the author have experimental data to explain this point?

References

The authors should cite pioneer studies that use cell extract system within GUVs

Methods

The calibration method to compare fluorescent intensities among samples (gain or exposure time, laser strength, and standard sample) is missing. Therefore, I cannot evaluate the reliability of the quantification data from confocal microscopy.

Purity of LactC2-eGFP

It seems too low purity to conclude something. I strongly suggest to add some data using more purified LactC2-eGFP through further purification steps.

Minor comments

L2, p2. Please refer some reference.

Figure 3. Typically, YFP is colored with yellow. Why the author colored it green? I suggest move Fig 3ab to supplementary, because this kind of work has been published elsewhere and are not their original.

Reference 51. The title is incorrect. Furthermore, the study did not show "liposome division".

Reviewer #3 (Remarks to the Author):

The manuscript by Blanken et al. demonstrated the possibility of membrane lipid synthesis within the encapsulated volume of lipid bilayer vesicles using cell-free expressed enzymes with transcriptional and metabolic regulation and subsequent incorporation of synthesized lipids into the vesicle

membranes. Four major types of lipids were detected in bulk and in confinement. Both quantitative and qualitative assays were developed to assess the yield and localization of the different lipids. As mentioned in the manuscript, the authors claim the major significance of the study as the ability of the presented system to synthesize a repertoire of membrane lipids which are the same as the ones used to make liposomes in the first place from a so called "mini-genome". By changing the concentrations of the added polymerases and altering the membrane lipid compositions, the final yield and type of lipids made could be controlled. Quantitative techniques like mass spectrometry and lipidomics were used to detect and quantify yield of reaction products at different stages of the enzymatic pathways. Two types of optical detection assays were also developed to monitor localization of synthesized lipids on liposomal membranes when expressed in confinement. This is indeed a major advance in the field and could be an impactful approach for possible growth and division of synthetic cell-like liposomes (although growth has not been demonstrated here). While expression of enzymes with cell-free expression systems have been attempted before with successful production of lipids like PA and PS, the work is novel in its use of a mini-genome-like plasmid with seven genes and a combination of acyl-CoA with different chain lengths to generate hybrid lipid mixtures in a one pot reaction. Overall, the manuscript is well written and organized. The work is quite relevant in the field of in vitro lipid synthesis and use of cell-free expression as a tool for reconstitution of enzymatic pathways and membrane proteins. There are however certain limitations of the presented approach that need to be elucidated to provide the full scope of the work to the readers. My major concerns are as follows.

1. The authors have stated that the primary reason for building a polycistronic plasmid with all the genes for the Kennedy pathway was to prevent heterogeneity in expression due to variation in plasmid copies when encapsulating multiple plasmids in liposomes. However, using the same promoter to express all the enzymes for the PE pathway doesn't allow control over the extent of expression of individual proteins. Since acyl-CoA and CTP are consistently used by several proteins of this pathway, there is a competition for these resources and having equal amounts of each protein will lead to non-optimal synthesis and low yield. Instead, using promoters with different strengths specific to the endogenous *E. coli* RNA polymerase can be beneficial to improve the rate and yield of the final products. Have the authors considered this option?

2. The plot in Fig. 2b shows similar yields of DOPE in the presence and absence of SP6 RNA polymerase for all values of DOPG concentration in the LUV membranes. The authors have mentioned that the PG synthesis pathway is hindered and results in low yield at high initial DOPG concentrations. However, at low DOPG concentrations, there seems to be no difference between the yield of DOPE with and without SP6 addition despite a significant amount of DOPG synthesis. Further discussion on this observation should be included. The authors should also comment that at 50% DOPG that the amount of DOPE synthesized actually dropped (or remained constant given the experimental error). The author showed only one representative experiment here and showed two repeats in Supp. Fig. 11, where only one of the two repeats had the same DOPG% and the other was very different (though a similar trend). It would be prudent to have the same experiments repeated three times to include for a main result in a manuscript.

3. There seems to be no difference between the number of vesicles with NBD signal on their membranes in the case with proteinase outside and the one with no proteinase (Supp Fig. 13). It has been previously shown that the rate of cell-free expression changes in confined volumes and varies with size of liposomes. But protein synthesis in bulk expression is much higher in yield and duration and so one might expect to see a much larger contrast in the NBD fluorescence with no proteinase in the outer CFE solution post rehydration of the lipid-coated beads. This is qualitatively shown in the bottom plot of Fig. 4b where the condition with no protease shows higher peaks for all the lipids species compared to the one with protease outside the liposomes. Was the outer CFE diluted and washed before incubation? I was also a bit surprised that the fluorescence intensity outside the liposomes in the 'no confinement' condition is not very high in Fig. 3a and 3b (the baseline between '1' and '2' in Figure 3b is not very large). I find the term 'no confinement' here may be confusing to the readers as I would think confinement here refers to encapsulation. I suggest the author change this

description. Figure 3b should display a y-axis range. I am not familiar with where the active sites are for these various lipid modifying enzymes. PS is found mostly in the inner leaflet of natural cells and the asymmetry is possible by specific flippases that exposes PS to the outer leaflet. Given the detection of PS is by purified LactC2-GFP added to the outside and from the schematic cartoons shown, the authors seem to suggest that PS is made on the outer leaflet. I was surprised that there was no discussion on the asymmetry (or symmetry) of the leaflets, as this may introduce caveat in the interpretation of the results.

4. The authors noted that the amount of de novo synthesized lipids incorporated in the membrane was not sufficient to observe liposome growth. The authors should show this quantified data. The example from Fig. 5f gives the impression that the liposome grew. This data is 'hidden' in Fig. 5g where the authors displayed liposome radius in color (which to me is not a good way to present this). Since it seems like the authors have looked at many individual liposomes over time, it should be relatively straightforward to measure the size at time 0 and compared with time at 10 hours for the same liposome. This should be displayed as paired data as a separate figure panel.

Minor comments:

1. Abstract: The first statement should be reworded. Lipid, nucleic acid, and proteins are classes of macromolecule. Metabolism is a process and not an 'ingredient'.
2. Page 2 line 25-27: 'In addition, liposome growth.....'. This is not a great sentence, please rephrase.
3. Page 3 line 26: Enzymes downstream 'of' the pathway are....
4. When calculating NBD enrichment, quantification here be biased by the fact that not all vesicles express enzymes as shown in earlier figures. What are the criteria for selecting GVs? There is some info in the methods on this, but it might helpful to include this in the main text for the readers. There appears to be a large fraction of liposomes that don't express lipids and the authors should discuss this observation beyond this being inherent to gene expression in cell-sized compartments. Is this also the case when there is no proteinase or DNase added outside?
5. Figure 3e. I found the color choices made the graph a bit difficult to read. It took me some time to decipher the different conditions. The magnified inset of d and e should have the y-axis scale labeled.
6. Page 6 line 30: '.....it relies also on the association-dissociation of PssA to the membrane'. Is there experimental evidence for this (from past literature or current work)?
7. Figure 3 caption: why is there heavy-isotope fraction of DOPE in the initial membrane?
8. Page 9 line 31: 'Concluding, it has been possible to.....'. This is a poor sentence, please rephrase.
9. Page 10 line 28: '..., whilst yielding a sufficiently....', no 'to'.
10. Have the authors investigated how much acyl-CoA is consumed over time? The authors commented on adding more than 100 μM of acyl-CoA is not possible due to low solubility. Given the groups expertise in mass spec, it might be possible to look at the consumption of acyl-CoA. I am still surprised with the addition of synthesized lipids that there is no apparent increase in liposome size.
11. There are a large number of supplemental figures. While they generally help make the case, I question whether 1) they all need to be shown, and 2) whether they support the claims made. While it is ok sometimes to show data from $n = 1$ experiments, I don't think this provides the level of rigorousness that would be needed, even if they are shown in supplemental figures. For instance, Supp. Fig. 4 is unimpressive, and one should really include the comparison of pGEMM7 on the same

graph as a proper control. Supp. Fig. 8 needs to have other controls and repeated. Supp. Fig. 15 showed examples of linescan for 1 and 12% PS that aren't very different. There is also an issue with experimental design for Supp. Fig. 15e that looks to have missing data point and the legend is quite strange. Supp. Fig. 16c has no values and units on y-axis and shows a $n = 1$ experiment. Supp. Fig. 17: The authors claimed that compartmentalized PS synthesis results in a higher coefficient of variation in LacC2 signaling than when PS is directly included in vesicle membrane. The significance of this finding is unclear and not well placed in the context of this work. There are missing error bars for the eGFP sample. Why do 0% PS liposomes have LactC2 binding? Supp. Fig. 18d: The quantification shown here does not corroborate with the image shown in 18a. The fraction of LactC2-mCherry bound liposomes seems to be far less than 40%. Supp. Fig. 20 also has $n = 1$, even though the result is quite striking.

12. Supp. Fig. 5: There is discrepancy between the caption and what is indicated in the main text (Page 5 line 16). Is this using *E. coli* cell-free expression or by PURE?

13. Supp. Fig. 21: Is there a reason why fadD expression reduces DOPE synthesis in the presence of O-CoA?

Reviewer #4 (Remarks to the Author):

Review of NCOMMS-20-01773-T

The authors present a manuscript which describes the cell-free expression of phospholipid-producing enzymes in liposomes. They can show that these enzymes are functional and convert lipid precursors into a variety of lipids, including those which constitute the liposome. The authors could confirm that this minimal cell systems can regulate a balanced production of different lipids via metabolic feedback. I was asked to comment on the technical aspects of the mass spectrometry experiments of this study. The successful expression of the enzymes in the liposomes was confirmed using a targeted proteomics approach, by tryptic digest of the peptides followed by identification via MS/MS. EF-Tu is a constant component of the PURE system and was used as a global internal standard. The proteins are partially identified by not many peptides (for example only one in the case of CdsA (P0ABG1)). But as these are membrane proteins, a lower coverage than for soluble proteins is to be expected, and the identification justified.

The lipids synthesized in the liposomes were analyzed by LC-MS and could be distinguished from the originally present lipids as the precursors were labeled with ^{13}C -G3P, resulting in a 3Da mass shift. Conversion of integrated peaks for the lipid fragments was into lipid concentrations was based on linear calibration curves obtained from dilution series of standards.

One concern: The authors state "the average integrated counts of two injections was determined." Does that mean two injections of the same sample? To get reliable data, I would expect data from three independent experiments, as was done for the proteomics analysis.

Apart from the point mentioned above, the proteomics and lipidomics experiments are performed in a technically sound manner, including the appropriate controls and support the conclusions of the authors, which present a study, which I consider overall a quite exciting step towards use of minimal cells for investigation of separated aspects in cell function.

Rebuttal letter for:

Manuscript: "Genetically controlled membrane synthesis in liposomes" by Blanken et al.

Article reference: NCOMMS-20-01773-T

We are grateful to the Referees for their careful review and constructive comments that helped us improve the manuscript. We have taken all the comments into consideration and have made appropriate changes in the revised manuscript. The Referee reports appear below in *blue text* and our point-by-point responses are in *black text*. Changes in the manuscript (main text or supplementary information) are highlighted in *red text*. All figure numbers, references and page numbers mentioned in this letter refer to their numbering in the revised manuscript.

Reviewer #1:

In this manuscript the Danelon laboratory does a good job of developing a cell-free system for the synthesis of lipids. Such work is of clear importance to the synthesis of synthetic cells and may have additional biotechnological applications. The experiments are generally well designed and the data are convincing.

This group did publish previously in PLOS ONE similar work that demonstrated the synthesis of lipid. It is not immediately clear what the difference is with this submitted manuscript. The authors should do a better job in indicating what the new advance is.

I think this work is important and will be used by several laboratories as they try to build artificial cells.

Reply We thank the reviewer for acknowledging the importance of our work. The present study provides important conceptual advances and new experimental results that clearly go beyond previous reports, including our work published in PLOS ONE (Scott et al., 2016). Compared to our previous article, we have encoded the seven lipid synthesizing enzymes on a single DNA template, called pGEMM7. This allowed us to take two essential steps towards an autonomously growing minimal cell: first, we could investigate both genetic and metabolic regulation of the lipid synthesis pathway. Secondly, encoding all genes on a single piece of DNA was paramount to reconstitute the whole Kennedy pathway inside giant vesicles. This stands in contrast with our previous results,

where the full set of enzymes was expressed outside large unilamellar vesicles. In our earlier report, only the first two enzymes in the pathway could be expressed from the inside of liposomes, but the lipid product was an intermediate species that was not a constituent of the parental vesicle. Hence, compartmentalization of the entire phospholipid metabolic pathway inside vesicles, as demonstrated here, represents a key milestone in the construction of an autonomous artificial cell.

Furthermore, the LC-MS lipidomics approach that we employed in our previous study is here complemented with a targeted proteomics analysis, as well as two novel fluorescence microscopy assays for direct visualization of membrane synthesis at the single-vesicle level.

We concur with the reviewer that it should be made clearer for the reader what the innovative aspects and new advance of the present study are. Therefore, we have modified the text on page 3, lines 2-9: “**However, the output phospholipid PA was not part of the original membrane composition and the PA detection method was not compatible with single vesicle resolution [23]. Regeneration of the main constituents of the liposome membrane obligates the reconstitution of five additional headgroup-modifying enzymes, which together with PlsB and PlsC form the Kennedy metabolic pathway that produces phosphatidylethanolamine (PE) and phosphatidylglycerol (PG), the most abundant lipids in the *E. coli* membranes. Although the unregulated expression of the Kennedy pathway enzymes was enabled from the outside of liposomes [23], in vesiculo synthesis of membrane-forming lipids with controlled molecular ratios remains a challenge.**”

I do, however, have suggestions that the authors may wish to consider during revisions.

1. I do not believe that the authors ever demonstrated that the membrane proteins that were cell-free expressed actually partitioned to the membrane. Functional activity is certainly important, but it would be nice to demonstrate the localization of these proteins since that may aid future work.

Reply In our previous work (Scott et al., 2016), membrane incorporation of PlsB and PlsC was demonstrated using a liposome floatation assay combined with co-translational fluorescence labelling of the proteins (Fig. 1b in ref. 23). Moreover, we found that formation of LPA and PA, the two reaction products of PlsB and PlsC, respectively, were consistently one order of magnitude higher when the proteins were expressed in the presence of phospholipid vesicles (Fig. 2b in ref. 23). We concluded that membrane localization greatly enhances enzymatic activity for these proteins. In the present study, we showed in **Fig. 2** that the lipid composition of the vesicles (% of PG content) modulates lipid production, presumably acting as an allosteric regulator of PssA and PlsB. These

findings can best be explained if the involved enzymes associate with the membrane. In addition, the transmembrane nature of some enzymes, like CdsA, implies that the active protein partitions into the lipid bilayer.

In future works, it would be insightful to determine the localization of the different proteins in the membrane. Specifically, it would be relevant to determine whether all proteins are evenly distributed in the vesicle membrane or if they form functional clusters. We think, however, that such investigations go beyond the scope of the present study.

2. Liposome growth was not demonstrated but could have easily been observed with FRET labeled lipids. This would strengthen the manuscript considerably since the goal appears to be growth.

Reply Both phospholipid synthesis and membrane incorporation of the output phospholipids have been unambiguously shown in this manuscript. Together, these two processes represent membrane growth. Due to the limited amount of acyl-CoA precursor that could be supplied, the total amount of synthesized lipids only resulted in a growth of ~5% of the membrane area. This value is modest relative to the 100% membrane expansion that is necessary for a full proliferation cycle, which is our long-term goal. Nonetheless, liposome growth was demonstrated.

We agree that FRET-based methods would be a powerful tool to quantify liposome growth, especially in a regime where the magnitude of membrane expansion is small. This technique has been widely used in liposome research and we are familiar with it. The outcome of such an assay will be an accurate assessment of the increase of the membrane area, which we know will range between 0% and ~10%. The knowledge of the 'exact' % increase has little added value, in our opinion, given that our next goal is to achieve substantial growth that is directly observable by standard microscopy methods, i.e. without FRET. Therefore, the quantitative results from FRET assays will not really help us improve the vesicle growth efficiency. For these reasons, we decided to not conduct FRET experiments in the context of this study.

3. The figures frequently lack statistical analysis and an indication of the number of replicates.

Reply For all figures in the main text, number of replicates is noted in the caption and, when appropriate, the individual data points are displayed in the graph. In the revised manuscript, **Fig. 2b** has been replotted to include data previously shown in Supp. Fig. 11, hence displaying the three repeats in the main text. In **Fig. 5d,g**, datasets from multiple experiments have been combined; both

the number of experiments and the number of analysed liposomes are explicitly stated in the figure caption.

We added statistical analysis of **Fig. 5d** and **Fig. 5e**. The following sentences have been added to the corresponding figure captions.

Caption Fig. 5d: “Membrane-localised GFP fluorescent intensity is significantly higher when both DNA and oleoyl-CoA were present compared to negative controls, with $p < 0.0001$ (two-sample Welch’s t-test).”

Caption Fig. 5e: “The amount of PS-enriched liposomes is significantly higher when both DNA and oleoyl-CoA were present compared to the two negative controls, with $p < 0.04$ (two-sample Welch’s t-test).”

Sometimes the units of the axes are missing.

Reply We reviewed all figures, including in the supplemental information, and added units on the axes of **Figs. 3b,d,e**, and **Supp. Fig. 16c**.

4. The first two sentences of the abstract are probably a little too strong regarding what is needed for life.

Reply We agree that the first sentence might be too strong if one considers primitive ‘living’ systems, i.e. intermediates in the evolutionary pathway from lifeless molecules to contemporary organisms. However, lipid membranes, nucleic acids, proteins and metabolism are universal ingredients of modern-day life. To not exclude the possibility that protocells were of a different nature and to avoid a disputation about the essence of life, we have rewritten the first sentences to restrict our definition to the current biology: “Lipid membranes, nucleic acids, proteins, and metabolism are essential for modern cellular life. Synthetic systems emulating the fundamental properties of living cells must therefore be built upon these functional elements.”

5. The introduction begins by stating that life is an individual construct, but that is again, likely too strong. Evolution gives rise to populations. I know what the authors mean, but more care should be taken in how these ideas are expressed.

Reply It was not our intention to start with a provocative statement about the nature of life. It would certainly be more accurate to say that the simplest manifestation of life is in the form of individual cellular entities. To avoid ambiguity, and realising that it does not support a key message, we decided to remove this sentence.

6. Last sentence of the introduction: "Our results provide experimental evidence for DNA-encoded homeostatic growth of a liposome-based artificial cell." But the authors do not demonstrate growth in this manuscript.

Reply By demonstrating both the synthesis and membrane incorporation of bilayer-forming phospholipids, we have demonstrated vesicle growth. The fact that the increase in liposome size is below the diffraction limit of light does not mean that no growth has taken place. See also our reply to point 2.

7. It is not clear why data was split between figure 2 and Supplementary Figure 11. The two should be combined together.

Reply We have replotted **Fig. 2** accordingly and we removed Supp. Fig. 11.

8. The degradation of material external to the liposomes was either with proteinase K or DNase I. Is there a reason why both weren't used together?

Reply The methods are mutually exclusive since proteinase K will degrade DNase I. Using either one of them was sufficient to shut down gene expression outside liposomes.

9. Line 5, page 10: It is unclear what is meant by "hidden in ensemble measurements."

Reply We modified the sentence as: "While LC-MS methods provide sensitive detection of multiple lipid species in a liposome population, information about lipid composition at the single vesicle level is lost due to vesicle solubilisation."

10. Supplementary Figure 4: Why are the lipid concentrations in arbitrary units?

Reply The y axis represents integrated peak intensity of ¹³C-labelled lipids. It correlates with lipid concentration but it is not a direct measure of concentration. No absolute quantification was performed for this experiment, since we were interested here in relative amounts of lipids. In the revised figure, the data have been normalized to the signal of ¹²C-DOPE to account for differences in MS sensitivity, and the y-axis label has been modified accordingly. Note that we have included data with pGEMM7 at the request of Reviewer #3.

11. Supplementary Figure 4: The legend states "We hypothesized that this was caused by transcription termination read-through by the T7 RNAP, transcribing downstream PG synthesis genes even if those are under control of an orthogonal promoter." One way to avoid this problem is to clone the different modules in different directions.

Reply This was also our reasoning and we accordingly modified pGEMM6 by cloning the modules in different directions, giving rise to the plasmid pGEMM7 that has been extensively used in this research. We have added the following sentence to the figure caption to clarify this point: **"This finding prompted us to design the new construct pGEMM7, where the two sets of PE and PG pathway genes were cloned in opposite directions to ensure full orthogonality."**

12. Supplementary Figure 8: without error bars, the data are hard to interpret

Reply The experiment has been performed only once. Therefore, no error bars are provided. When replacing *pgpA* with *pgpC*, the presence of synthesized DPPG could unambiguously be determined, demonstrating the activity of PgpC. We decided to remove this figure as it should be repeated for higher scientific accuracy and because it does not support a main claim.

13. Figure 3b: The difference between the dashed lines is confusing and not enough to appreciate the logic of the experiment.

Reply We have replotted the graph with a logarithmic y-axis to more clearly display the decrease of external fluorescence upon addition of proteinase K or DNase.

Reviewer #2:

In this study, Blanken and colleagues constructed a single plasmid encoding seven genes involved in lipid synthesis and showed protein synthesis using a reconstituted transcription/translation system (PURE system) from the information. The results showed that the 7 protein involved in lipid synthesis was successfully synthesized in the liposomes, and indicated the synthesis of the objective lipids.

The direction in which this research aims is certainly important in the future of synthetic biology.

Reply We thank the reviewer for acknowledging the significance of the research area.

However, questions remain regarding the achievement, novelty of the thesis, and conceptual advances. The authors conclude in the title "membrane synthesis", but no evidence has been presented beyond lipid synthesis as the author mentioned that they failed to observe an increase of liposomes sizes by their system.

Reply The fact that we achieved membrane synthesis is simply not questionable. We are surprised that the Reviewer missed this important aspect of our work. The results of both fluorescence-based assays (**Figs. 4** and **5**, and related supporting figures) unambiguously demonstrate that - at least a fraction of - the internally synthesized phospholipids are effectively incorporated to the membrane, yielding vesicle growth.

We've been very explicit throughout the article about the current limitations of our system toward effective growth from an internal phospholipid factory. As stated on page 14, lines 3-5, "the amount of de novo synthesized lipids incorporated in the membrane was not sufficient for directly observing liposome growth under an optical microscope." Further, on page 16, lines 31-32, we wrote "no visible membrane or volume expansion could unambiguously be measured by optical microscopy". The new panel **Fig. 5h** is also meant to illustrate that no 'visible' increase of the vesicle radius was observed within the spatial resolution of the microscope. However, vesicle growth, be it modest, is obviously demonstrated, as commented above.

This is in contrast to reference 19 that shows this point.

Reply In ref. 19 (now ref. 21 in the revised manuscript), the authors have reconstituted detergent-solubilized enzymes into small vesicles. They showed by LC-MS phospholipid production starting from the fatty acid precursor. Data from an R18-self-quenching assay and from electron microscopy

indicate membrane expansion, although no statistical analysis of the liposome size is provided. Our experimental framework is radically different: the enzymes are expressed from their gene, biosynthesis of proteins and lipids is compartmentalized inside large/giant vesicles, to mention the most striking differences. De novo membrane synthesis and vesicle growth was more convincingly shown by the group of Devaraj (refs. 13,14,51). But, there too, the experimental framework differs from ours, and the respective advantages and shortcomings are different.

Regarding novelty, previous studies have already developed technologies to combine seven or more genes into one plasmid.

Reply We never claimed that the construction of a seven-gene plasmid is new. A plethora of multi-gene constructs have already been assembled in vitro and in vivo. The entire synthetic biology field is routinely generating large DNA constructs, including synthetic chromosomes. There are very little examples though of >2-gene constructs employed in cell-free gene expression. The most prominent achievement is that of Shepherd et al. 2017 (in *Nucleic Acid Res.*), who reported a 30-cistron plasmid. What is truly novel in our work is the design and the use (not the construction per se) of a multi-cistron DNA that supports functional expression and regulation of a metabolic pathway in a cell-free system and, to a larger extent, within liposomes.

Expression and insertion to the membrane of the 7 proteins involved in lipid synthesis have been shown in their previous study. Although the only novel point of the manuscript is the expression of the 7 proteins in liposomes, there are many reports that showed normal protein synthesis (including membrane proteins as the author already reported) via the PURE system within liposomes.

Reply PURE system expression of (membrane) proteins inside liposomes has already been achieved, including by our group. However, in the vast majority of the studies, expression of a single or two genes is reported. Several technical challenges must be overcome to express an entire catalytic pathway of several enzymes within liposomes. This is precisely what we have accomplished for the first time in this study, including also the implementation of regulatory mechanisms.

We object to the statement that “expression of the 7 proteins in liposomes [...] is the only novel point of the manuscript”. It is an unfair assessment of our work, in sharp contrast with the opinion of the other reviewers. Below, we summarize the key novel results.

We have previously shown the functional reconstitution of the seven lipid synthesis enzymes in the PURE system (Scott et al., 2016). Importantly, this was realized from individual DNA constructs, expressed at the outside of small unilamellar vesicles and only ensemble LC-MS data were provided (no single-vesicle resolution). Here, aided by the construction of the pGEMM7 **mini-genome**, we are able to recapitulate the full enzymatic pathway **inside giant liposomes**. By design, we equipped liposomes with **genetic control** over cell-free expression of the branched enzymatic pathway (**Fig. 1**). All these aspects represent major conceptual advances towards an autonomously growing minimal cell. In addition, we show for the first time (i) in vitro reconstitution of the membrane-mediated control of PE vs. PG synthesis (**Fig. 2**), (ii) the one-pot synthesis of six lipid species (**Fig. 3**), (iii) the use of NBD-labelled precursors for cell-free synthesis of artificial NBD-labelled phospholipids (**Fig. 4**), (iv) the use of the lactC2-eGFP probe (**Fig. 5**). The latter two assays provide the first evidence of lipid synthesis and membrane incorporation on the **single vesicle level** by fluorescence microscopy and reveal **liposome-to-liposome variability**. Finally, the lactC2 probe allows us to measure the incorporation **kinetics** of newly synthesized PS (**Fig. 5**).

These novel points are therefore not only technical, but conceptual in nature and we are surprised to notice that they have been overlooked by the Reviewer.

Due to the levels of achievement they set and lack of significant novelty and conceptual advances, I do not support the current manuscript for publishing in Nature Communication.

Reply We hope that our point-by-point responses provide the necessary clarifications and a better positioning of our study.

Other major comments

L2, p3. Figure 5 shows synthesis of PS within GUVs. However, PS is also not part of the original membrane composition of E. coli system.

Reply PS is an intermediate phospholipid species in the *E. coli* membrane. Accumulating PS by silencing expression of Psd was a necessary modification of the system to make use of the LactC2 probe. We have also demonstrated lipid synthesis and membrane incorporation of PE and PG by employing NBD-labelled lipid precursors (**Fig. 4**), which did not require a modification of the enzymatic pathway. Moreover, the LC-MS data reported in **Figs. 2** and **3** show successful production of the two output lipids PE and PG when the pathway is fully activated.

L9, p3. No evidence of the growth of liposome is shown in this manuscript.

Reply This is not correct. Both phospholipid synthesis and membrane incorporation of the resultant phospholipids have been unambiguously shown in this manuscript. These two processes together represent membrane growth. Please, refer to page 7 of this letter for a more elaborate argumentation.

L27, p5. I think normalization using a different peptide fragment is a non-standard method in MS-based quantification. Do the authors have a reference or data that support the feasibility of the normalization?

Reply Using the same peptide fragment is only required for absolute quantification, which is not the purpose of this experiment. Reviewer #4, who was specifically asked to review our mass spectrometry methods, states that “the proteomics and lipidomics experiments are performed in a technically sound manner, including the appropriate controls and support the conclusions of the authors”. Reviewer #4 also explicitly acknowledges the use of EF-Tu as an internal global standard.

Figure 3&4. I wonder why the protease or DNase outside liposomes affect protein expression within liposomes. Does the author have experimental data to explain this point?

Reply In both **Figs. 3** and **4**, we observe a decrease in total lipid synthesis activity when gene expression is exclusively confined to the interior of liposomes by addition of protease or DNase. We do not attribute this decrease to any deleterious effect of protease or DNase on internal gene expression, as suggested by the Reviewer. Inhibiting protein synthesis outside liposomes strongly reduces the total volume in which lipid production takes place. If one takes into account this drastic reduction in volume, the decrease in enzymatic activity measured at the population level is actually surprisingly low. On page 11, lines 30-33, we added: “the moderate increase (~50%) of the fraction of NBD-enriched liposomes when omitting the proteinase K (**Fig. 4d**) might be explained by an enhancement of enzymatic activity in liposome-confined reactions, as suggested above for lipid production at the population level (**Fig. 3d,e**).”

References

The authors should cite pioneer studies that use cell extract system within GUVs

We agree with the Reviewer's suggestion and cited the pioneer studies of Noireaux et al. 2004 and Nomura et al. 2003 in the revised manuscript as new refs. 4 and 5.

[4] Nomura, S. et al. Gene expression within cell-sized lipid vesicles. *ChemBioChem* **4**, 1172-1175 (2003).

[5] Noireaux, V. & Libchaber, A. A vesicle bioreactor as a step toward an artificial cell assembly. *Proc. Natl. Acad. Sci. U.S.A* **101**, 17669-74 (2004).

Methods

The calibration method to compare fluorescent intensities among samples (gain or exposure time, laser strength, and standard sample) is missing. Therefore, I cannot evaluate the reliability of the quantification data from confocal microscopy.

Reply Within data sets, identical imaging settings were always used, which allows for direct comparison of fluorescence intensities. For clarity, we have explicitly stated this in the revised manuscript, in the Methods section. Moreover, in **Figs. 4d** and **5e**, we report the percentage of enriched liposomes, which is based on a threshold determined for each individual sample, as explained in the corresponding figure captions. These results can therefore be compared irrespective of imaging settings. Specifically, we added:

On page 25, lines 31-32: "Within data sets, identical imaging settings were used."

On page 9, line 8: "[...] the same size and were acquired with identical imaging settings."

On page 15, line 28: "Identical imaging settings were used for all acquired data."

Purity of LactC2-eGFP

It seems too low purity to conclude something. I strongly suggest to add some data using more purified LactC2-eGFP through further purification steps.

Reply Firstly, we quantified the level of nonspecific binding at different concentrations of purified LactC2-eGFP and we used reference liposomes with different amounts of PS for calibration (**Supp. Fig. 15**). From these experiments, we found the optimal concentration of LactC2-eGFP to specifically report PS exposed to the vesicle membrane. Note that the concentration of LactC2-eGFP was also determined by absorbance measurement of eGFP. Moreover, our results were confirmed using different batches of purified LactC2-eGFP and a different protein fusion (LactC2-mCherry, **Supp. Figs.**

17 and **18**). Therefore, we are confident that our conclusions are fully valid. Secondly, we think the purity of LactC2-eGFP as shown in the lane 'BE' (buffer exchange) of **Supp. Fig. 12a**, is sufficient, although a relative low amount of protein was loaded onto this gel.

Regarding the presence of two to three prominent bands of the LactC2-mCherry fusion protein observed in **Supp. Fig. 12b**, they do not reflect a low purity. Instead, they correspond to different denaturation forms of LactC2-mCherry that migrate differently on gel. This observation has already been reported in the case of another mCherry fusion protein [new Supp. Ref. 3]. In their study, Mestrom et al. found that a stabilized form of the fusion protein was resistant to at least 2% SDS and ran at a lower apparent molecular weight by PAGE. This more native form of the fusion protein would correspond to one of the lower bands of LactC2-mCherry marked with a circle symbol in **Supp. Fig. 12b**. Further purification of LactC2-mCherry by gel filtration (Sephacryl S200 16/60 HiPrep mounted on an ÄKTA Pure system, both from GE Healthcare) failed to eliminate the extra bands visible by Coomassie staining (see the gel below, upper panel), indicating that they all correspond to the same protein.

Figure caption: Analysis of gel filtration fractions of purified LactC2-mCherry on 4-12% Bis-Tris SDS-PAGE gels. Samples from different purification steps were loaded on the gels. The upper panel shows a fluorescence image of the gels to localize the folded, active mCherry. The bottom panel shows the same gels stained with Coomassie to visualize the total protein content. The red boxes indicate the fluorescent band of the native mCherry fusion protein. The other two prominent bands above and below very likely correspond to differently denatured states of the mCherry, similar to what Mestrom *et al.* observed [3]. CFE, cell-free extract; HisTrap frac, pooled HisTrap fractions corresponding to the starting material for gel filtration; M, PageRuler™ Plus Prestained protein ladder (ThermoFisher); void, void volume of column; A8 to C1, gel filtration fractions.

Additionally, we analyzed these LactC2-mCherry samples using anion exchange chromatography (MonoQ 4.6/100 PE column on an ÄKTA Pure system, GE Healthcare). Isolated samples were treated with 2 M urea and 95 °C for different periods of time, and were loaded on a 4-12% Bis-Tris SDS gel ran in MES buffer (see figure below). The two prominent bands corresponding to fully and partially denatured states of the mCherry fusion protein can be observed. The middle one that is visible in the fluorescence channel disappeared after heat treatment, further indicating that it corresponds to a more folded, native form.

Concluding, the prominent bands observed in the lane of the purified LactC2-mCherry all correspond to the same protein and the level of purity is sufficient for utilization as a reporter probe for PS lipids.

Figure caption: SDS-PAGE analysis of LactC2-mCherry samples subjected to different denaturing conditions. The left panel shows a fluorescence image, where only the folded, active mCherry can be seen. The right panel shows the same gel stained with Coomassie to visualize the total protein content. AF, affinity chromatography; GL, gel filtration chromatography; AIX, anion exchange chromatography. The time in minutes indicates the duration of the treatment at 95 °C in Laemmli sample buffer supplemented with 10 mM DTT. The ‘yes’ or ‘no’ refers to the addition, or not, of 2 M urea in the sample loading buffer. The blue arrow points to the fully denatured LactC2-mCherry. The magenta arrow indicates the native protein, of which the mCherry fluorescence disappears after heat treatment. The ‘M’ lane is the ladder in kDa. The lane ‘+’ corresponds to gel filtration sample in Bolt 4xLDS sample buffer (ThermoFisher) incubated for 10 minutes at 75 °C.

In the revised manuscript, the above section about LactC2-mCherry was inserted as **Supplementary Note 3** and the two figures as **Supp. Figs. 13** and **14**.

New Supp. Ref. 3: Mestrom, L., Marsden, S.R., Dieters, M., Achterberg, P., Stolk, L., Bento, I., Hanefeld, U., Hagedoorn, P.L. Artificial fusion of mCherry enhances trehalose transferase solubility and stability. *Appl. Environ. Microbiol.* **85**, e03084-18 (2019).

Minor comments

L2, p2. Please refer some reference.

Reply Reviewer #1 commented that the statement was too strong and we decided to remove this sentence.

Figure 3. Typically, YFP is colored with yellow. Why the author colored it green?

Reply The current magenta-green color scheme is internationally agreed upon since it gives the highest contrast for both regular and color-blind people.

I suggest move Fig 3ab to supplementary, because this kind of work has been published elsewhere and are not their original.

Reply We think that these pictures are important at this point of the manuscript to show the features of liposome samples (size, morphology, fraction of expressing vesicles). Moreover, a comparison of the reporter gene expression upon addition of proteinase K or DNase has not been published elsewhere.

Reference 51. The title is incorrect. Furthermore, the study did not show “liposome division”.

Reply The reference has been modified, thank you. Liposome division by cell-free expressed proteins has not been demonstrated yet. Reference 51 (old numbering) is most appropriate when discussing the ‘liposome division’ functional module as it represents a major advance to reconstitute (part of) the bacterial divisome by cell-free gene expression inside liposomes. We also cited the recent work of Ueda and colleagues (new ref. 57) in the revised manuscript.

Corrected ref. 56: Godino, E. et al. De novo synthesized Min proteins drive oscillatory liposome deformation and regulate FtsA-FtsZ cytoskeletal patterns. *Nat. Commun.* **10**, 4969 (2019).

New ref. 57: Furusato, T. et al. De novo synthesis of basal bacterial cell division proteins FtsZ, FtsA, and ZipA inside giant vesicles. *ACS Synth. Biol.* **7**, 953–961 (2018).

Reviewer #3 (Remarks to the Author):

The manuscript by Blanken et al. demonstrated the possibility of membrane lipid synthesis within the encapsulated volume of lipid bilayer vesicles using cell-free expressed enzymes with transcriptional and metabolic regulation and subsequent incorporation of synthesized lipids into the vesicle membranes. Four major types of lipids were detected in bulk and in confinement. Both quantitative and qualitative assays were developed to assess the yield and localization of the different lipids. As mentioned in the manuscript, the authors claim the major significance of the study as the ability of the presented system to synthesize a repertoire of membrane lipids which are the same as the ones used to make liposomes in the first place from a so called “mini-genome”. By changing the concentrations of the added polymerases and altering the membrane lipid compositions, the final yield and type of lipids made could be controlled. Quantitative techniques like mass spectrometry and lipidomics were used to detect and quantify yield of reaction products at different stages of the enzymatic pathways. Two types of optical detection assays were also developed to monitor localization of synthesized lipids on liposomal membranes when expressed in confinement.

This is indeed a major advance in the field and could be an impactful approach for possible growth and division of synthetic cell-like liposomes (although growth has not been demonstrated here).

Reply We thank the reviewer for her/his accurate summary of our work and for recognizing that it represents a major advance in the field. We highly appreciate the thorough review and extensive comments that helped us clarify several points in the revised manuscript.

While expression of enzymes with cell-free expression systems have been attempted before with successful production of lipids like PA and PS, the work is novel in its use of a mini-genome-like plasmid with seven genes and a combination of acyl-CoA with different chain lengths to generate hybrid lipid mixtures in a one pot reaction.

Overall, the manuscript is well written and organized. The work is quite relevant in the field of in vitro lipid synthesis and use of cell-free expression as a tool for reconstitution of enzymatic pathways and membrane proteins.

Reply Thank you for pointing out the relevant and novel aspects of our studies.

There are however certain limitations of the presented approach that need to be elucidated to provide the full scope of the work to the readers. My major concerns are as follows.

1. The authors have stated that the primary reason for building a polycistronic plasmid with all the genes for the Kennedy pathway was to prevent heterogeneity in expression due to variation in plasmid copies when encapsulating multiple plasmids in liposomes.

*However, using the same promotor to express all the enzymes for the PE pathway doesn't allow control over the extent of expression of individual proteins. Since acyl-CoA and CTP are consistently used by several proteins of this pathway, there is a competition for these resources and having equal amounts of each protein will lead to non-optimal synthesis and low yield. Instead, using promoters with different strengths specific to the endogenous *E. coli* RNA polymerase can be beneficial to improve the rate and yield of the final products. Have the authors considered this option?*

Reply As a proof-of-concept of genetic control on cell-free phospholipid synthesis, we have opted to orthogonally regulate the two branches of the Kennedy pathway with T7 and SP6 RNAP. Because the T7 RNAP is endogenous to the PURE system, it is therefore the most logical starting point when designing a transcriptionally regulated pathway. We concur with the reviewer that employing the native *E. coli* RNA polymerase along with sigma factors would expand the capabilities to modulate transcription of individual genes and fine-tune the level of individual proteins. To emphasize the benefit of this approach in future work, we added the following text in the Discussion (page 17, lines 4-7): “Alternatively, employing the native *E. coli* RNA polymerase along with sigma factors would expand the capabilities to regulate transcription of individual genes and fine-tune the level of individual proteins [48,49]. This could in turn ameliorate the production rate and yield of the output lipids.”

Additional citations:

[48] Shin, J. & Noireaux, V. An *E. coli* cell-free expression toolbox: application to synthetic gene circuits and artificial cells. *ACS Synth. Biol.* **1**, 29–41 (2011).

[49] Maddalena, L.L., Niederholtmeyer, H., Turtola, M., Swank, Z.N., Belogurov, G.A. & Maerkl, S.J. GreA and GreB enhance expression of *Escherichia coli* RNA polymerase promoters in a reconstituted transcription-translation system. *ACS Synth Biol.* **5**, 929–935 (2016).

2. The plot in Fig. 2b shows similar yields of DOPE in the presence and absence of SP6 RNA polymerase for all values of DOPG concentration in the LUV membranes. The authors have mentioned that the PG synthesis pathway is hindered and results in low yield at high initial DOPG

concentrations. However, at low DOPG concentrations, there seems to be no difference between the yield of DOPE with and without SP6 addition despite a significant amount of DOPG synthesis. Further discussion on this observation should be included.

Reply PE synthesis is downregulated by the high PE amount in the membrane, irrespective of the presence of a competing pathway. We stated in the main text “Interestingly, PE synthesis was reduced at low PG content, independent of the expression of the PG-synthesizing pathway branch (Fig. 2c, Supp. Fig. 11). This result indicates that the regulatory mechanism is not solely driven by competition between the two pathway branches but it relies also on the association-dissociation of PssA to the membrane”. We think that the level of discussion is sufficient given the claims we make and the reported data sets. We’d rather avoid speculating about putative mechanisms that lack in vivo confirmation anyways.

The authors should also comment that at 50% DOPG that the amount of DOPE synthesized actually dropped (or remained constant given the experimental error). The author showed only one representative experiment here and showed two repeats in Supp. Fig. 11, where only one of the two repeats had the same DOPG% and the other was very different (though a similar trend). It would be prudent to have the same experiments repeated three times to include for a main result in a manuscript.

Reply We have replotted the main text figure to include all three experiments (new **Fig. 2b-d**). The fact that in one experiment the input PG% values are slightly different is not important as we pooled all data without calculating mean \pm s.d.v values. Note that PG% values are measured parameters and are therefore also subject to experimental variability. We’ve been careful to base our statements on obvious trends of the data: (i) the concentration of synthesized PE increases with the proportion of input PG with and without SP6 RNAP (new **Fig. 2b**), (ii) the concentration of synthesized PG decreases as the % of input PG increases when the PG branch is activated (new **Fig. 2c**), and (iii) the total amount of synthesized lipids (PE+PG) increases as the % of input PG increases (new **Fig. 2c**). Our data do not provide evidences that the amount of synthesized PE drops or remains constant at 50% input PG. The aggregated data clarify this point.

3. There seems to be no difference between the number of vesicles with NBD signal on their membranes in the case with proteinase outside and the one with no proteinase (Supp Fig. 13). It has been previously shown that the rate of cell-free expression changes in confined volumes and varies

with size of liposomes. But protein synthesis in bulk expression is much higher in yield and duration and so one might expect to see a much larger contrast in the NBD fluorescence with no proteinase in the outer CFE solution post rehydration of the lipid-coated beads. This is qualitatively shown in the bottom plot of Fig. 4b where the condition with no protease shows higher peaks for all the lipids species compared to the one with protease outside the liposomes.

Reply We would like to start by clarifying some of the reviewer's statements. Protein synthesis with PURE_{frex2.0} in bulk expression is not much higher in duration and concentration compared to in-liposome reactions; see our study cited as ref 38, where a similar protocol for gene expressing liposomes was applied.

In Supp. Fig. 13 (now **Supp. Fig. 11**), the number of liposomes in the field of view is not large enough to conclude that NBD-positive liposomes is the same in both conditions, especially if one considers background signal from the NBD-substrate and liposome-to-liposome variability. **Fig. 4d** provides a quantitative analysis of such experiments and it clearly shows a higher percentage of NBD-enriched vesicles in the absence of proteinase K. Moreover, the reviewer expects a stronger NBD signal on the membrane of vesicles in the sample with no proteinase, which is a reasonable prediction. However, re-plotting the data in **Fig. 4d** as histograms of NBD fluorescence intensity for the different conditions tested (figure below, each panel represents an independent data set), it can be seen that single-vesicle NBD signals are not so much different with no protease and with protease outside. The vertical dashed line indicates the intensity threshold above which vesicles are considered enriched in NBD-labelled phospholipids.

In the bottom graph of **Fig. 4b**, the NBD fluorescence intensity measured by HPLC corresponds to the integrated intensity for each separated NBD-labelled species in the vesicle population. The higher peaks observed in the sample without proteinase result from the higher percentage of liposomes that are NBD-enriched, not from the higher NBD fluorescence on individual liposomes.

It can be noticed that the fraction of NBD-enriched liposomes is higher with no proteinase compared to proteinase added outside (**Fig. 4d**), but not largely higher as one could expect given the much larger volume outside liposomes. Similarly, **Fig. 3d,e** shows that the concentration of synthesized lipids is higher with no proteinase, but not largely higher, as measured by LC-MS for six different lipid species. The latter observation has been discussed on page 8 lines 13-15. We attribute this effect to an enhancement of enzymatic activity inside the confined environment of the GUV, in line with the reviewer's remark that "the rate of cell-free expression changes in confined volumes". Two citations supporting this hypothesis have been added (i.e., refs. 4 and 38). The same hypothetical scenario applies for **Fig. 4d**, but we forgot to specify this. In the revised manuscript, on page 11, lines 30-33, we clarified this point by adding: "In addition, the moderate increase (~50%) of the fraction of NBD-enriched liposomes when omitting the proteinase K (**Fig. 4d**) might be explained by an enhancement of enzymatic activity in liposome-confined reactions, as suggested above for lipid production at the population level (**Fig. 3d,e**)."

Was the outer CFE diluted and washed before incubation?

Reply On page 11, lines 22-23, we wrote: "After pGEMM7 expression, the liposomes were diluted to reduce the membrane signal coming from NBD-palmitoyl-CoA and NBD-LPA." As detailed in the Methods section, the sample of surface-immobilized liposomes was washed three times with buffer E to remove non-reacted NBD-palmitoyl-CoA. For HPLC (**Fig. 4b**), the (NBD-labelled) lipid fraction was extracted using methanol, without any washing steps. The corresponding protocol was also mentioned in the Methods section. For clarity, we have modified the caption of **Fig. 4d** on page 13, lines 1-2, as: "... in **b** (bottom). The samples were washed three times to remove non-reacted NBD-palmitoyl-CoA."

I was also a bit surprised that the fluorescence intensity outside the liposomes in the 'no confinement' condition is not very high in Fig. 3a and 3b (the baseline between '1' and '2' in Figure 3b is not very large).

Reply In this particular experiment, liposomes were diluted (2 μ L in 7.5 μ L total) to reduce their surface density and aid visualization. This was not clearly stated in the paper. Therefore, we added this information in the caption of **Fig. 3a**. We have previously observed that YFP expression inside individual GUVs can be significantly higher than in the surrounding bulk (see ref. 38) The fluorescence contrast is more pronounced here as the sample was diluted. Such an observation was

also reported in the pioneer work of Nomura et al. (new ref. 4). These results further support a scenario where liposome-compartmentalized gene expression is enhanced compared to bulk reactions.

On page 9, lines 6-7, we added: “Liposomes were diluted (2 μ L in 7.5 μ L total) to reduce their surface density and aid visualisation.”

I find the term ‘no confinement’ here may be confusing to the readers as I would think confinement here refers to encapsulation. I suggest the author change this description.

Reply We agree. We changed the term ‘no confinement’ in **Fig. 3a** and **Supp. Fig. 10**, and ‘without confinement’ in the caption of **Fig. 3a** by ‘No protease/DNase’.

Figure 3b should display a y-axis range.

Reply We changed it.

I am not familiar with where the active sites are for these various lipid modifying enzymes. PS is found mostly in the inner leaflet of natural cells and the asymmetry is possible by specific flippases that exposes PS to the outer leaflet. Given the detection of PS is by purified LactC2-GFP added to the outside and from the schematic cartoons shown, the authors seem to suggest that PS is made on the outer leaflet.

Reply In **Fig. 1a**, the cartoon illustrates lipid production occurring from the outside of the liposomes because the PURE system was mixed with preformed SUVs. In **Figs. 3c, 4a** and **5a**, the cartoons illustrate enzyme and lipid production from the inside of GUVs, with one of the precursors, the acyl-CoA, being supplied from the outside of the liposomes. In the latter configuration, we presume that a fraction of PS partitions in the outer leaflet where it is exposed to the LactC2 probe. However, this does not necessarily imply that PS is synthesized on the outer leaflet and we did not intend to suggest so. Given that the PS-producing enzyme, PssA, is a membrane-associated protein that exists in both the membrane-bound and free states (**Fig. 2a**), it is more likely that synthesized PS is released from the active site into the inner leaflet. Subsequently, PS may flip to the outer leaflet, a process that is not energetically favourable and requires assistance of specialized enzymes in vivo. However, the artificial bilayer of our liposomes is not as rigid and is more prone to transient defects

compared to cell membranes (refs. 38 and 42). Therefore, membrane dynamic processes such as lipid flip-flop and translocation of small molecules may be less impaired in liposomes than in cells, facilitating partitioning of PS in the outer leaflet, and possibly of other synthesized lipids too. These differences in the membrane properties may also explain why the bilayer-spanning enzymes of the Kennedy pathway spontaneously insert in the membrane without an active machinery. Note also that we cannot rule out the possibility that LactC2-GFP permeates across the membrane and binds PS exposed to the lumen. We've been prudent to not comment on this aspect given the level of speculation. However, in the light of the reviewer's comment, we acknowledge that it requires some discussion which we included in the revised manuscript (see next comment).

I was surprised that there was no discussion on the asymmetry (or symmetry) of the leaflets, as this may introduce caveat in the interpretation of the results.

Reply Following on the previous comment, the question of the partitioning of the synthesized lipids in the inner and outer leaflets of the liposomes is highly relevant and deserves more attention in the Discussion. We've been prudent in the interpretation of our results. For instance, we assigned LactC2 membrane recruitment to a 'PS enrichment', without specifying its location. **Fig. 5a** illustrates the scenario where the externally supplied LactC2 binds to PS present in the outer leaflet. In our opinion, this is the most probable scenario. Nonetheless, we decided to provide a more complete discussion on page 16, lines 4-12.

“What is the distribution of internally produced lipids in the bilayer? Phosphatidylserine is likely synthesized on the inner leaflet of the liposome membrane [41]. Nevertheless, synthesized PS is detected on the outer leaflet, where it is exposed to the LactC2-eGFP probe. Flipping of phospholipids is not energetically favourable and requires assistance of specialized enzymes in vivo. However, the artificial bilayer of our liposomes is not as rigid as the bacterial cell membrane and is more prone to transient defects [38,42]. Therefore, membrane dynamic processes, such as lipid flip-flop and translocation of small molecules, may be less impaired in liposomes, facilitating partitioning of PS (and possibly of other synthesized lipids too) in the outer leaflet. Although the possibility that LactC2-GFP permeates across the membrane and binds PS exposed to the lumen cannot be excluded, this process is severely hindered by the bulky fusion protein.”

New ref. 41: Larson, T., & Dowhan, W. Ribosomal-associated phosphatidylserine synthetase from *Escherichia coli*: purification by substrate-specific elution from phosphocellulose using cytidine 5'-diphospho-1,2-diacyl-sn-glycerol. *Biochemistry* **15**, 5212-5218 (1976).

4. The authors noted that the amount of de novo synthesized lipids incorporated in the membrane was not sufficient to observe liposome growth. The authors should show this quantified data. The example from Fig. 5f gives the impression that the liposome grew. This data is 'hidden' in Fig. 5g where the authors displayed liposome radius in color (which to me is not a good way to present this). Since it seems like the authors have looked at many individual liposomes over time, it should be relatively straightforward to measure the size at time 0 and compared with time at 10 hours for the same liposome. This should be displayed as paired data as a separate figure panel.

Reply We agree that our statement that “no visible membrane or volume expansion could unambiguously be measured by optical microscopy” should be supported by quantified data. As suggested, we have included a new panel in figure 5 (**Fig. 5h**), which displays the change in apparent radius between the start and end points of the kinetics reported in **Fig. 5g**. No significant increase in apparent radius could be observed. Moreover, we have extended the analysis shown in **Supp. Fig. 19** to investigate the relationship between change in apparent radius and reaction rate, plateau time, and liposome size (**Supp. Fig. 19e,f**). No clear correlations were observed.

We added in the caption of **Fig. 5**: “**h**, Probability distributions of the apparent radius change for the liposomes analysed in panel (**g**). The apparent radius change was determined by calculating the difference between the apparent radius at 0 h and 16 h. The apparent radius was calculated based on the perimeter of the liposome selection marker under the assumption of a perfect circle. A bin width of 250 nm was chosen.”

Minor comments:

1. Abstract: The first statement should be reworded. Lipid, nucleic acid, and proteins are classes of macromolecule. Metabolism is a process and not an 'ingredient'.

Reply We have rewritten the sentence as “Lipid membranes, nucleic acids, proteins, and metabolism are essential for modern cellular life.”

2. Page 2 line 25-27: 'In addition, liposome growth.....'. This is not a great sentence, please rephrase.

Reply We rephrased as: “To establish a link between the lipid compartment and its internal content, liposome growth could be made conditional to encapsulated nucleic acids [12,16] or catalysts [17].”

3. Page 3 line 26: Enzymes downstream 'of' the pathway are....

Reply We changed as “Enzymes downstream in the pathway”.

4. When calculating NBD enrichment, quantification here be biased by the fact that not all vesicles express enzymes as shown in earlier figures. What are the criteria for selecting GVs? There is some info in the methods on this, but it might helpful to include this in the main text for the readers.

Reply The Methods section has been moved from the supplemental information to the main text. Moreover, we have added the following sentence on page 12, lines 21-22: “Liposomes were selected for unilamellarity based on the Texas Red channel.”

There appears to be a large fraction of liposomes that don't express lipids and the authors should discuss this observation beyond this being inherent to gene expression in cell-sized compartments. Is this also the case when there is no proteinase or DNase added outside?

Reply Non-expressing liposomes are also observed in the absence of proteinase or DNase (**Fig. 3a**) in the case of YFP expression. **Supp. Fig. 11a** (right panel) shows liposomes with no (at least no more than in negative control samples) NBD signal when gene expression was enabled inside and outside the vesicles. We complemented the discussion on page 16, lines 26-29, to account for other mechanisms that may contribute to the observed heterogeneity in lipid expression: “In the present experiments, other sources of heterogeneity in lipid enrichment may also contribute, such as a variability in the adsorption of acyl-CoA among liposomes upon resuspension of the precursor film. Investigating the mechanisms leading to phenotypic differences will be important to further optimise the chain of reactions from genes to output lipids.”

5. Figure 3e. I found the color choices made the graph a bit difficult to read. It took me some time to decipher the different conditions.

Reply We took a color map that is designed to provide as much contrast as possible, also for color blind people (<https://cran.r-project.org/web/packages/viridis/vignettes/intro-to-viridis.html>). We tried several options to plot the data from six different lipids in eight different conditions with three repeats, and we think this is the best we can do.

The magnified inset of d and e should have the y-axis scale labeled.

Reply The insets of **Fig. 3d,e** have been modified accordingly.

6. Page 6 line 30: ‘.....it relies also on the association-dissociation of PssA to the membrane’. Is there experimental evidence for this (from past literature or current work)?

Reply The transient association of PssA with the membrane is conditional to the presence of anionic lipids, as described in refs. 30-32. We have cited again these papers on page 6, line 32.

7. Figure 3 caption: why is there heavy-isotope fraction of DOPE in the initial membrane?

Reply In nature, various isotopes coexist for almost all elements. For example, 1.1% of carbon in nature is ^{13}C , having a mass 1 Da higher than the most prevalent ^{12}C . Because of this, molecules that are slightly heavier than calculated from the standard elemental weights are a common occurrence. This becomes obvious when trying to distinguish between regular DOPE and ^{13}C -labelled DOPE, which is only 3 Da heavier. Apparently, ~1% of naturally occurring DOPE is 3 Da heavier than the atomic weight of 743.5 Da, and is therefore detected when scanning for the isotopically labelled ^{13}C -DOPE. For DOPG, two molecules of ^{13}C -labelled G3P are incorporated, causing a mass difference of 6 Da between the light DOPG and our reaction product. Apparently, DOPG with a 6-Da mass shift does not occur in nature in detectable amounts.

For clarity, we slightly rephrased the corresponding text on page 10, lines 6-8, as: “A small amount of DOPE was measured in samples where no acyl-CoA was supplied. This represents the naturally occurring heavy-isotope fraction of the DOPE contained in the initial liposome membrane.”

8. Page 9 line 31: ‘Concluding, it has been possible to.....’. This is a poor sentence, please rephrase.

Reply We rephrased as: “Concluding, it has been possible to selectively produce up to six different lipid species (DOPE, DOPG, DPPE, DPPG, POPE, POPG) with a one-pot reaction coupling gene expression and phospholipid synthesis within cell-sized liposomes.”

9. Page 10 line 28: ‘..., whilst yielding a sufficiently....’, no ‘to’.

Reply We changed it.

10. Have the authors investigated how much acyl-CoA is consumed over time? The authors commented on adding more than 100 μM of acyl-CoA is not possible due to low solubility. Given the groups expertise in mass spec, it might be possible to look at the consumption of acyl-CoA.

Reply We found that transitions of 514.6 \rightarrow 78.9 and 1030.3 \rightarrow 407.9 gave clear peaks for oleoyl-CoA. We provide below a chromatogram displaying the total ion count (black, major peak) and the two transitions (lower peaks), measured on a standard of pure oleoyl-CoA. The retention time is 1.4 min, indicating limited adsorption on the column. We have performed a repeat of the experiment shown in **Fig. 2b**, where we investigated the amount of oleoyl-CoA after synthesis. Compared to a sample without lipid synthesis, we found that between 1% (0% PG, +SP6 RNAP) and 22% (50% PG, +SP6 RNAP) of input oleoyl-CoA remained unreacted. This result indicates that a significant fraction of oleoyl-CoA was consumed.

I am still surprised with the addition of synthesized lipids that there is no apparent increase in liposome size.

Reply When preparing GUVs as shown in **Fig. 5**, we add 1 mg of lipid-coated beads per 1 μL of PURE system solution. Every gram of beads contains 3.33 mg of lipids, or equivalently, 4.24 μmol of lipids. So, our liposome solution has an initial lipid concentration of 3.33 $\mu\text{g}/\mu\text{L}$ or 4.24 $\text{nmol}/\mu\text{L}$, corresponding to a lipid concentration of 4.24 mM. Prior to the microscopy experiments shown in

Fig. 5, this liposome solution was subsequently diluted 3.75 times, giving a final concentration of lipids of 1.13 mM.

In the ideal case, where all oleoyl-CoA (100 μ M) is consumed, 50 μ M of lipids are synthesized. This represents an increase in lipid concentration of $(1.13 \text{ mM} + 0.05 \text{ mM})/1.13 \text{ mM} = 1.045$, so 4.5 %. Assuming that i) lipid synthesis is homogeneously distributed, ii) all supplied lipids assemble into liposomes, and iii) all lipid species in the membrane have the same molecular surface area, the area of liposomes is also increased by 4.5%. Therefore, the diameter, which scales with the square root of the area, increases by 2.2%. For an average liposome diameter of 4 μ m, this means a diameter increase of 88 nm, which is well below the detection limit of our light microscope.

Let's assume now that only 50% of the liposomes synthesize lipids and that all supplied oleoyl-CoA feeds the active liposomes. Then, a surface area increase of 9% is expected, so a diameter increase of 4.4%. For our 4 μ m diameter liposomes, this results in a growth of 176 nm, still less than one pixel of 250 nm.

For the sake of the argument, one could calculate the increase in lipid concentration needed to grow a liposome of 4 μ m diameter to 4.5 μ m diameter, i.e. a detectable growth. Such vesicle growth corresponds to a liposome surface area increase of 26%. Under the assumption that the complete lipid film forms liposomes that synthesize lipids, this means 300 μ M of phospholipids need to be synthesized, requiring 600 μ M oleoyl-CoA. At such a high concentration, oleoyl-CoA would precipitate in the PURE system reaction buffer.

11. There are a large number of supplemental figures. While they generally help make the case, I question whether 1) they all need to be shown, and 2) whether they support the claims made. While it is ok sometimes to show data from $n = 1$ experiments, I don't think this provides the level of rigor that would be needed, even if they are shown in supplemental figures.

Reply We generally agree with the Reviewer's comment. We decided to remove the supplementary figures that report preliminary data from a single experiment and that do not directly provide extended data from main text figures. These are former Supp. Figs. 8, 20 and 21.

For instance, Supp. Fig. 4 is unimpressive, and one should really include the comparison of pGEMM7 on the same graph as a proper control.

Reply Although the data may not look ‘impressive’, they prompted us to rethink and change our plasmid design. We therefore think it is useful to show it. We agree that including pGEMM7 data will aid interpretation of the result, and have therefore included the 0 AU and 4 AU SP6 RNAP data from **Fig. 1c** in the revised **Supp. Fig. 4**. We modified the caption as: “**b**, Synthesis of phospholipids from pGEMM6 (data with pGEMM7 are appended for comparison), [...] Bars are average values from two independent repeats (three with pGEMM7), each represented by a different symbol.”

Supp. Fig. 8 needs to have other controls and repeated.

Reply We removed this figure.

Supp. Fig. 15 showed examples of linescan for 1 and 12% PS that aren't very different.

Reply We agree with this observation. As can be seen at the population level, the mean GFP rim intensity shown in the box plot aren't that different for 1% and 12% PS. This observation, combined with similar results for LactC2-mCherry (**Supp. Fig. 18c**), indicates that the relationship between the concentration of PS and LactC2-probe fluorescence intensity is nonlinear. In addition, it can be seen in **Supp. Fig. 15c** that the histograms of LactC2-eGFP fluorescence at 1% and 12% PS significantly overlap. We've been prudent to report the fraction of PS-enriched liposomes instead of absolute fluorescence of LactC2-eGFP/mCherry as a measure of PS synthesis.

To bring this observation to the attention of the reader, we added in the caption of **Supp. Fig. 15c,d**: “The distributions of LactC2-eGFP fluorescence at 1% and 12% PS significantly overlap. [...] The relationship between LactC2-eGFP fluorescence intensity and PS concentration is nonlinear.”

There is also an issue with experimental design for Supp. Fig. 15e that looks to have missing data point and the legend is quite strange.

Reply No data points are missing from the graph, but it is true that the legend layout was confusing. We have clarified it.

Supp. Fig. 16c has no values and units on y-axis and shows a n = 1 experiment.

We have added y-axis labels.

Some of the experiments presented in **Supp. Fig. 16c** have been repeated multiple times, e.g. with linear pGEMM (+EcoR1). Herein, we only report data from a single experiment, where all conditions have been tested at once to minimize technical variability. The obtained results are very clear: PS only accumulates when the *psd* gene is inactivated, and no major difference is measured between the three strategies to silence gene expression outside liposomes. Further quantitation of the data was not necessary.

Supp. Fig. 17: The authors claimed that compartmentalized PS synthesis results in a higher coefficient of variation in LacC2 signaling than when PS is directly included in vesicle membrane. The significance of this finding is unclear and not well placed in the context of this work.

Reply This finding is significant because it reveals the heterogeneous nature of lipid production, which is an important property of the vesicle population.

This statistical analysis is relevant due to the inherent variability of LactC2-probe fluorescence intensity even in liposomes with predetermined amounts of purified PS (**Supp. Fig. 15c,d**). To be more explicit on the relevance of the coefficient of variation, we modified the text on page 13, lines 22-26 : “A wide distribution of eGFP intensity values in PS-synthesizing liposomes was measured (Fig. 5d). The coefficient of variation is ~2-fold higher than in control samples with a predetermined fraction of PS (Supplementary Fig. 17). This result further supports the highly heterogeneous nature of liposome-encapsulated lipid synthesis.” In addition, we have added the following sentence in the caption of **Supp. Fig. 17**: “The coefficient of variation provides a good measure of the excess liposome-to-liposome heterogeneity of membrane-incorporated PS caused by cell-free protein and lipid synthesis.”

We have clarified the significance of this finding by adding the following text in the Discussion on page 16, lines 26-29: “In the present experiments, other sources of heterogeneity in lipid enrichment may also contribute, such as a variability in the adsorption of acyl-CoA among liposomes upon resuspension of the precursor film. Investigating the mechanisms leading to phenotypic differences will be important to further optimise the chain of reactions from genes to output lipids.”

There are missing error bars for the eGFP sample.

Reply We wrote in the caption: “No error bar indicates that the experiment was performed once.” We modified as: “Calibration experiments with LactC2-eGFP have been performed once”. Two to

three independent experiments were already conducted with LactC2-mCherry. We considered that one experiment with LactC2-eGFP was sufficient here because the two probes behave similarly, as shown in **Supp. Figs. 15** and **18**.

Why do 0% PS liposomes have LactC2 binding?

Reply Unspecific binding of membrane probes is a commonly observed phenomenon and is more prevalent at higher LactC2 concentrations. The working concentration of LactC2 probe should be chosen to minimize unspecific binding and provide a high signal-to-noise ratio. Thus, it must be high enough to ensure good signal, but not too high to specifically report on PS. In the caption of **Supp. Fig. 15e**, we wrote: “Clear non-specific binding of the LactC2-eGFP probe to the membrane can be observed at high concentration. To ensure PS-binding specificity, a working concentration of LactC2-eGFP of 150 nM was employed in all measurements with expressed pGEMM7”, addressing this issue.

Supp. Fig. 18d: The quantification shown here does not corroborate with the image shown in 18a. The fraction of LactC2-mCherry bound liposomes seems to be far less than 40%.

Reply We have replaced the panels shown in **Supp. Fig. 18a,b** to display a more representative field of view.

Supp. Fig. 20 also has $n = 1$, even though the result is quite striking.

Reply We decided to remove this figure, see previous point on page 27 of this letter.

*12. Supp. Fig. 5: There is discrepancy between the caption and what is indicated in the main text (Page 5 line 16). Is this using *E. coli* cell-free expression or by PURE?*

Reply The experiments displayed in the original Supp. Fig. 5 were performed using an *E. coli* cell lysate, unlike what we wrote in the main text. Thank you for pointing out this discrepancy. For consistency, we repeated these experiments with PURE $_{flex}$ 2.0 and reported the data in the revised **Supp. Fig. 5**. As expected, results are similar as those obtained in the *E. coli* lysate, confirming orthogonality between the T7 and SP6 promoters. We have modified the caption and Methods section accordingly.

13. Supp. Fig. 21: Is there a reason why fadD expression reduces DOPE synthesis in the presence of O-CoA?

Reply Only one or two experiments were conducted, which is not enough to draw solid conclusions. We decided to remove this figure, see previous point on page 27 of this letter.

We appreciate the reviewer's efforts to provide extensive comments and constructive feedback.

Reviewer #4 (Remarks to the Author):

Review of NCOMMS-20-01773-T

The authors present a manuscript which describes the cell-free expression of phospholipid-producing enzymes in liposomes. They can show that these enzymes are functional and convert lipid precursors into a variety of lipids, including those which constitute the liposome. The authors could confirm that this minimal cell systems can regulate a balanced production of different lipids via metabolic feedback.

I was asked to comment on the technical aspects of the mass spectrometry experiments of this study. The successful expression of the enzymes in the liposomes was confirmed using a targeted proteomics approach, by tryptic digest of the peptides followed by identification via MS/MS. EF-Tu is a constant component of the PURE system and was used as a global internal standard. The proteins are partially identified by not many peptides (for example only one in the case of CdsA (P0ABG1)). But as these are membrane proteins, a lower coverage than for soluble proteins is to be expected, and the identification justified.

The lipids synthesized in the liposomes were analyzed by LC-MS and could be distinguished from the originally present lipids as the precursors were labeled with ^{13}C -G3P, resulting in a 3Da mass shift. Conversion of integrated peaks for the lipid fragments was into lipid concentrations was based on linear calibration curves obtained from dilution series of standards.

Reply We thank the reviewer for the detailed summary and careful evaluation of our mass spectrometry methods.

One concern: The authors state "the average integrated counts of two injections was determined." Does that mean two injections of the same sample? To get reliable data, I would expect data from three independent experiments, as was done for the proteomics analysis.

Reply We refer here to two injections of the same sample. For each condition, three biological repeats (independent experiments) were performed, as mentioned in the captions of **Figs. 1,2 and 3**. Elaborating, for every biological replicate, we have performed two injections, the average of which was used in further analysis. We opted for this approach since we noticed that carry-over from previous injections could cause biased lipid concentration measurements. By twice injecting the samples in random order, we have attempted to mitigate this effect. Furthermore, we sometimes observed a minor decrease in MS sensitivity during the course of our measurements. Randomly

injecting twice, and measuring a standard curve before and after the samples for quantification, was applied to average out bias introduced by this.

Apart from the point mentioned above, the proteomics and lipidomics experiments are performed in a technically sound manner, including the appropriate controls and support the conclusions of the authors, which present a study, which I consider overall a quite exciting step towards use of minimal cells for investigation of separated aspects in cell function.

Reply Thank you for the positive comment.

Reviewer #1 (Remarks to the Author):

I am satisfied with the response to my comments and the changes made to the manuscript. In my opinion, the manuscript is suitable for publication.

Reviewer #2 (Remarks to the Author):

I think significance of the paper depends on the conclusion of whether membrane synthesis is actually achieved by proteins expressed within liposomes. Hence, this point should be supported by unquestionable data.

Major point: Membrane synthesis

Although the authors claimed that the method the authors use (assessments by binding of fluorescent proteins) is enough to show insertion of a significant fraction of lipids in the membranes, their optical microscope observation that does not have a spatial resolution ($\sim 1 \mu\text{m}$) cannot provide evidence on a lipid bilayer structure with a thickness of about 5 nm. Therefore, at this stage, it has not been shown whether the fluorescence is derived from a membrane (lipid bilayer embedded in liposomes) structure or not, and cannot deny a possibility of other lipid structures such as a micelle on membranes. LC-MS can demonstrate the existence of synthesized phospholipids, but not show the existence of bilayer as the authors also described in the response.

Reviewer 1 also suggested a method like FRET, but the authors declined the suggestion due to technical problems. An additional experiment to support the conclusion of membrane synthesis is necessary. The authors should consider other demonstration experiments to show membrane synthesis such as time-lapse observation using super-resolution microscopes or clear visualization of lipid structures by electron microscopes.

In any case, at the current manuscript, the synthesis of membranes are not "obviously demonstrated" as the author insisted in the responses.

Other points:

i) Regarding the response to Reviewer1

The authors argue that the expression inside liposomes is a remarkable point compared to previous studies (I agree about this point), but however, the authors simultaneously cite the results of expression from the outside in previous studies as an evidence to show the correct insertion of the membrane proteins synthesized. There is no guarantee that the same thing is happening in the cases of expression from outside and inside. This point also suggests that the author's conclusions are not fully supported.

ii) I suggest citing the original study in this field rather than the new references 4 & 5.
Journal of Bioscience and Bioengineering, Volume 92, Issue 6, 2001, Pages 590-593

Reviewer #3 (Remarks to the Author):

The revision by Blanken et al is well carried out and the modifications in the text made the content more clear (except for one place that I would suggest revising). Although there are questions that remain, I am in support of publication of this work in Nature Communications

With respect to the rebuttal letter, I do want to comment on the authors' response regarding demonstration of growth. All three technical content reviewers had commented on the insufficient demonstration of membrane growth. I think this was an important point in my first review, and I am

ambivalent about the authors' response to this point. I agree that phospholipid synthesis and membrane incorporation are unambiguously shown. I think these are the hard data and should be interpreted as such. Evidence for growth, as presented, was not strong enough. Growth is a physical effect and carries a different meaning. Taking a living cell as an example (human would be a good one as well). Lipid synthesis and incorporation would be necessary for growth, but not sufficient. There are processes in cells that remove membrane and there is also a requirement to add volume for growth. I think most people would agree that a cell without physical growth is not considered 'growing', even if lipid synthesis and incorporation are taking place. I am not necessarily saying this argument applies to the current in vitro system, but growth needs to be substantiated by data (e.g. like the FRET expt reviewer 1 suggested). I appreciate the back of envelope calculation by the authors to estimate consumption of oleoyl-CoA and used this to make the argument that vesicle growth would be sub-200 nm. One caveat of this analysis is that it depends on the initial lipid concentration. Even when I first read this manuscript, I was struck by the density of the liposomes appeared in the images. The author's ~5% lipid synthesis can be drastically increased by a lower initial lipid concentration (not oleoyl-CoA) – in essence by optimizing the starting conditions of the experiment. I am satisfied with the authors' added quantified data to compare liposome sizes. In my second read of the manuscript, I did not feel there was an emphasis on demonstrating membrane growth. While this remains a weakness of the work, I am fine with it.

My only other comment is related to the discussion on enzyme localization/membrane asymmetry. The additional text is fine with the exception of stating the possibility of LactC2-GFP permeating across the membrane. This is a pretty small protein and by stating this, what prevents other small proteins in the system from going in and out of the membrane? To me, suggesting this possibility is more problematic in data interpretation. While I do not think there is data on which leaflet PS is produced in and how it may end up in the outer leaflet, the result presented here is that PS is detected on the outer leaflet. I believe there is literature out there that show enzyme-free lipid flipper under certain conditions and I cannot recall the details to know if it is applicable here. I would suggest that the author remove the statement on the possibility of LactC2-GFP permeates across the membrane as I don't think it is a sensible possibility in this context.

Minor comments:

- pg 6, line 9: '...prior and posterior to data...' change to '....prior to and post data....'

- Figure 4d: I am still struck by how low the percentage of enriched liposomes are....less than 20% in the case of no protease compared to 5% in the case of protease.....something to think about.

Reviewer #4 (Remarks to the Author):

The authors met my concerns, I recommend publication of this interesting work.

Rebuttal letter for:

Manuscript: "Genetically controlled membrane synthesis in liposomes" by Blanken et al.

Article reference: NCOMMS-20-01773A

We are grateful to the Referees for their second review and constructive comments. The Referee reports appear below in *blue text* and our point-by-point responses are in *black text*. Changes in the manuscript are highlighted in *red text*.

REVIEWERS' COMMENTS:

Reviewer #1 (Remarks to the Author):

I am satisfied with the response to my comments and the changes made to the manuscript. In my opinion, the manuscript is suitable for publication.

Reply Thank you!

Reviewer #2 (Remarks to the Author):

I think significance of the paper depends on the conclusion of whether membrane synthesis is actually achieved by proteins expressed within liposomes. Hence, this point should be supported by unquestionable data.

Reply As stated by Reviewer #3, 'phospholipid synthesis and membrane incorporation are unambiguously shown'.

Major point: Membrane synthesis

Although the authors claimed that the method the authors use (assessments by binding of fluorescent proteins) is enough to show insertion of a significant fraction of lipids in the membranes, their optical microscope observation that does not have a spatial resolution (~1 μm) cannot provide evidence on a lipid bilayer structure with a thickness of about 5 nm. Therefore, at this stage, it has not been shown whether the fluorescence is derived from a membrane (lipid bilayer embedded in liposomes) structure or not, and cannot deny a possibility of other lipid structures such as a micelle on membranes. LC-MS can demonstrate the existence of synthesized phospholipids, but not show the existence of bilayer as the authors also described in the response.

Reply First, the spatial resolution of our microscope is ~250 nm, not 1 μm . Second, PS lipids are not stable in micellar structures. Because of their cylinder shape, they will preferentially form bilayers. In

the absence of more solid arguments supporting an alternative mechanism, our observations are best explained by membrane synthesis through enzymes expressed within liposomes.

As an additional point of argumentation, the LactC2-eGFP probe has been tested on liposomes where PS was included as part of the liposome membrane (Supplementary Fig. 15). This resulted in similar colocalization characteristics as when PS was synthesized, which supports our claim of membrane incorporation. Furthermore, the NBD experiments in the present study and the results reported for synthesized DPPA in Fig. 4 and Supplementary Fig. 3 of our previous research (ref. 23) support membrane incorporation.

Reviewer 1 also suggested a method like FRET, but the authors declined the suggestion due to technical problems. An additional experiment to support the conclusion of membrane synthesis is necessary. The authors should consider other demonstration experiments to show membrane synthesis such as time-lapse observation using super-resolution microscopes or clear visualization of lipid structures by electron microscopes.

Reply This is incorrect, we have not declined to perform this experiment because of 'technical problems'. In our reply to Reviewer #1, we wrote: "The knowledge of the 'exact' % increase has little added value."

Transmission electron microscopy is not a suitable technique for >400-nm liposomes and the crowding protein environment (in particular ribosomes and some translation factors) will complicate direct visualization of lipid structures. Super-resolution fluorescence imaging will suffer from thermal membrane fluctuations. We are not saying that these two techniques are not relevant for future experiments, but it will be extremely challenging to reliably measure such a minute change of membrane surface area or liposome diameter. Anyways, as mentioned above, we do not think that additional experiments are necessary to strengthen our conclusion about membrane synthesis.

In any case, at the current manuscript, the synthesis of membranes are not "obviously demonstrated" as the author insisted in the responses.

Other points:

i) Regarding the response to Reviewer1

The authors argue that the expression inside liposomes is a remarkable point compared to previous studies (I agree about this point), but however, the authors simultaneously cite the results of expression from the outside in previous studies as an evidence to show the correct insertion of the membrane proteins synthesized. There is no guarantee that the same thing is happening in the cases of expression from outside and inside. This point also suggests that the author's conclusions are not fully supported.

Reply Membrane insertion or association of the involved enzymes is a prerequisite for their functions. Phospholipid synthesis is demonstrated; hence, functional incorporation of the membrane proteins is indirectly demonstrated too. We do not claim that *all* expressed proteins are active and correctly inserted. However, a fraction definitely is, both when gene expression occurs inside or outside vesicles.

A possible difference in the efficiency of membrane incorporation between inside and outside expression, could result from a difference in membrane curvature between the LUVs (for which direct evidence of membrane protein insertion was provided, see ref. 23) and giant vesicles, respectively used for outside and inside expression. However, for such large liposomes we expect curvature effects to be negligible.

ii) I suggest citing the original study in this field rather than the new references 4 & 5. Journal of Bioscience and Bioengineering, Volume 92, Issue 6, 2001, Pages 590-593

Reply We added the suggested reference as new Ref. 2, but we kept references 4 & 5 as they both provide more robust data on cell-free gene expression in liposomes.

Reviewer #3 (Remarks to the Author):

The revision by Blanken et al is well carried out and the modifications in the text made the content more clear (except for one place that I would suggest revising). Although there are questions that remain, I am in support of publication of this work in Nature Communications.

Reply We thank the reviewer for his/her positive opinion and for providing thoughtful suggestions.

With respect to the rebuttal letter, I do want to comment on the authors' response regarding demonstration of growth. All three technical content reviewers had commented on the insufficient demonstration of membrane growth. I think this was an important point in my first review, and I am ambivalent about the authors' response to this point. I agree that phospholipid synthesis and membrane incorporation are unambiguously shown. I think these are the hard data and should be interpreted as such. Evidence for growth, as presented, was not strong enough. Growth is a physical effect and carries a different meaning. Taking a living cell as an example (human would be a good one as well). Lipid synthesis and incorporation would be necessary for growth, but not sufficient. There are processes in cells that remove membrane and there is also a requirement to add volume for growth. I think most people would agree that a cell without physical growth is not considered 'growing', even if lipid synthesis and incorporation are taking place. I am not necessarily saying this argument applies to the current in vitro system, but growth needs to be substantiated by data (e.g. like the FRET expt reviewer 1 suggested).

Reply We thank the reviewer for supporting our main claim that internal phospholipid synthesis and membrane incorporation are unambiguously shown. We understand the reviewer's viewpoint about the notion of 'physical growth' and decided to tone down the related claim in the revised version. Specifically, we modified the manuscript text below:

"Strategies are discussed to alleviate current limitations toward ~~more~~ effective liposome growth and self-reproduction." (page 1, line 25)

"Our results provide experimental evidence for DNA-encoded ~~homeostatic-growth membrane synthesis in~~ a liposome-based artificial cell." (page 3, line 11)

Synthesis of phospholipids from an internal machinery and their incorporation in the lipid bilayer inevitably results in liposome are essential steps toward physical growth. (page 17, line 4)

I appreciate the back of envelope calculation by the authors to estimate consumption of oleoyl-CoA and used this to make the argument that vesicle growth would be sub-200 nm. One caveat of this analysis is that it depends on the initial lipid concentration. Even when I first read this manuscript, I was struck by the density of the liposomes appeared in the images. The author's ~5% lipid synthesis can be drastically increased by a lower initial lipid concentration (not oleoyl-CoA) – in essence by optimizing the starting conditions of the experiment.

Reply As the reviewer indicates, the relative growth is not only dependent on the amount of lipid synthesized, but also on the initial amount of phospholipid present as part of the liposome membrane. Therefore, lowering the liposome concentration might boost the magnitude of growth. This, however, has two main drawbacks.

- Practically, the current high vesicle density allows us to obtain measurements of many liposomes within a single experiment with a relatively low-throughput method as confocal microscopy. This has been paramount in our analysis of, for example, PS-enrichment (Fig. 4d, Fig. 4e) and of the coefficient of variation (Supplementary Fig. 17). Moreover, it is the best way to address the high liposome-to-liposome variability. Our previous observation that a small subset of liposomes exhibits enhanced gene expression phenotypes (ref. 38) encouraged us to perform high-content liposome measurements, to not miss potentially relevant lipid synthesis phenotypes.
- Increasing the ratio of oleoyl-CoA:phospholipids by diluting liposomes before supplying oleoyl-CoA will be limited by the stability of liposomes because oleoyl-CoA acts as a detergent. A good compromise would be “to provide a continuous supply of low-concentration acyl-CoA”, as we suggested on page 17, line 21.

I am satisfied with the authors' added quantified data to compare liposome sizes. In my second read of the manuscript, I did not feel there was an emphasis on demonstrating membrane growth. While this remains a weakness of the work, I am fine with it.

Reply Thank you.

My only other comment is related to the discussion on enzyme localization/membrane asymmetry. The additional text is fine with the exception of stating the possibility of LactC2-GFP permeating across the membrane. This is a pretty small protein and by stating this, what prevents other small proteins in the system from going in and out of the membrane?

Reply Translocation of the LactC2-GFP across the liposome membrane is a plausible scenario that we cannot totally exclude. We have shown permeation of the externally supplied low-molecular weight Cy5 molecule (Ref. 38) and of GFP (Ref. 42). In the latter study, we extensively discussed possible mechanisms in the supplemental information, section “3. Possible mechanisms for membrane permeability” and supplementary figures 5 & 6. In particular, we hypothesized that different reactional components in vesicles are engaged in functional multiprotein/nucleic acid complexes that are incapable of passing through the bilayer defects, even if their molecular weight is

lower than that of GFP. This assumption is supported by experiments in continuous cell-free translation devices based on semipermeable ultrafiltration membrane with pore sizes (typical molecular mass cutoff is 30 kDa) theoretically large enough for the diffusion of smaller translation factors [*]. Although the vesicle experiments in Ref. 42 were conducted with a different lipid composition and a slightly different protocol for liposome formation, we believe that such considerations might be valid in the present study too.

[*] a) A.S. Spirin, V.I. Baranov, L.A. Ryabova, S.Y. Ovodov, Y.B. Alakhov, *Science* **1988**, *242*, 1162; b) M. B. Iskakova, W. Szaflarski, M. Dreyfus, J. Remme, K. H. Nierhaus, *Nucleic Acids Res.* **2006**, *34*, e135.

To me, suggesting this possibility is more problematic in data interpretation. While I do not think there is data on which leaflet PS is produced in and how it may end up in the outer leaflet, the result presented here is that PS is detected on the outer leaflet. I believe there is literature out there that show enzyme-free lipid flipper under certain conditions and I cannot recall the details to know if it is applicable here. I would suggest that the author remove the statement on the possibility of LactC2-GFP permeates across the membrane as I don't think it is a sensible possibility in this context.

Reply We have been prudent in our data interpretation. Even if LactC2-GFP were to cross the liposome membrane and bind to PS on the inner leaflet, none of our conclusions would be affected. The main claim we make is that PS is produced from internally synthesized enzymes and that it incorporates in the liposome membrane, a process that can be imaged at the single-vesicle level with the LactC2-GFP probe. For the reasons mentioned above, and to give the reader all the elements to understand the experimental outcomes, we would like to keep the sentence on the possibility that LactC2-GFP may cross the membrane.

Minor comments:

- pg 6, line 9: '...prior and posterior to data...' change to '....prior to and post data....'

Reply We have modified the main text accordingly.

- Figure 4d: I am still struck by how low the percentage of enriched liposomes are....less than 20% in the case of no protease compared to 5% in the case of protease.....something to think about.

Reply We agree that the difference in percentage of enriched liposomes as determined with the NBD assay is not spectacular. This is a result of the rather high background signal due to incorporation of unreacted NBD-p-CoA in the membrane. Even after washing, membranes in the negative control exhibit significant NBD fluorescence (~500 a.u., Supplementary Fig. 11). This necessitates the use of a rather stringent threshold to exclude false positives. This has the detrimental effect of likely underestimating the 'real' number of PS-enriched liposomes. Therefore, we complemented the NBD-based assay by the LactC2-based assay, which was found to be more specific and had lower background signal, at the concentration used. This allowed us to place a more accurate threshold, which revealed that in fact, >50% of liposomes are PS-enriched (Fig. 5e). This number is more in line with expectations based on previous work where roughly half of liposomes prepared in this manner were found to exhibit fluorescence as a result of the expression of yellow

fluorescent protein (Ref. 38).

Reviewer #4 (Remarks to the Author):

The authors met my concerns, I recommend publication of this interesting work.

Reply Thank you!